# Enhanced $CO_2$ Emissions Driven by Flooding in a Simulation of Palsa Degradation

Mélissa Laurent[1,6], Mackenzie R. Baysinger[1,6], Jörg Schaller[2], Matthias Lück[2], Mathias Hoffmann[2], Torben Windirsch[1,3,5], Ruth H. Ellerbrock[2], Jens Strauss[1], and Claire C. Treat[1,4]

[1]Alfred Wegener Institute Helmholtz Centre for Polar and Marine Research, Permafrost Research Section, Telegrafenberg, Potsdam, Germany
[2]Leibniz-Center for Agricultural Landscape Research (ZALF), Müncheberg, Germany
[3]Institute of Geosciences, University of Potsdam, Potsdam, Germany
[4]Center for Landscape Research in Sustainable Agricultural Futures, Aarhus University, Denmark
[5]Research Institute for Sustainability Helmholtz Centre Potsdam, Potsdam, Germany
[6]Institute of Geo-ecology, University of Potsdam, Potsdam, Germany

**Correspondence:** Mélissa Laurent (melissa.laurent@awi.de)

**Abstract.** Climate change is predicted to put most of the permafrost habitats in the discontinuous zone at risk of disappearing within the next few decades. On a decadal scale, abrupt permafrost thaw may result in larger C losses than gradual permafrost thaw, but drivers of C emissions are poorly understood. To investigate this, we measured C emissions from a palsa under simulated abrupt and gradual thaw scenarios. We continuously measured $CO_2$ and $CH_4$ emissions while deepening the permafrost table under flooded (abrupt) and non-flooded (gradual) conditions. Higher soil-moisture during permafrost thaw is commonly associated with decreasing $CO_2$ and increasing $CH_4$ emissions. Interestingly, our results showed consistent $CH_4$ uptake across all the cores from the palsa and a twofold increase in $CO_2$ emissions under abrupt thaw (flooded conditions). Peat quality analysis (FTIR) showed a higher degradation of C compounds at the permafrost table, likely due to the physical disruption of soil organic matter and the redox changes in the active layer caused by flooding. Averaged $CO_2$ emissions were significantly higher under abrupt thaw (150 mg-$CO_2$ $m^{-2}$ $h^{-1}$) compared to gradual thaw (70 mg-$CO_2$ $m^{-2}$ $h^{-1}$), with limited permafrost peat contribution. Conversely, permafrost thaw under gradual thaw contributed to a twofold increase in $CO_2$ emissions (57 to 98 mg-$CO_2$ $m^{-3}$ $h^{-1}$). Finally, $CO_2$ emissions increased with depth in saturated fens, suggesting that deep-rooted vegetation could be a transport pathway for $CO_2$ outside the growing season. Our findings underline the potential for increased $CO_2$ emissions during the transition to fen conditions under abrupt thaw scenarios and therefore the need for in-situ measurements.

# 1   Introduction

Peatlands are major terrestrial carbon (C) sinks composed of partially decomposed organic matter (OM). Northern peatlands store approximately $415 \pm 150$ Pg of C, with 46% (185 Pg C) located in permafrost-affected regions (Hugelius et al., 2020).
Decomposition of C is limited due to the frozen and saturated soil conditions that are characteristic of permafrost peatlands. These soil conditions promote C accumulation, while suspending the decomposition of lower layers of OM (Frolking and Roulet, 2007; Sannel and Kuhry, 2009; Turetsky et al., 2000). Permafrost-affected peatlands are particularly vulnerable to shifts in climatic conditions. In the discontinuous permafrost zone, permafrost forms as peat plateaus and palsa - round or elongated permafrost peat mound, maximum height of about 10 m, composed of frozen peat uplifted by ice lenses overlying mineral soil (Seppälä, 1986).

The Arctic is a dynamic and quickly changing biome, with a projected circum-polar temperature increase that is three to four times faster than the global average (Obu et al., 2019; Rantanen et al., 2022; Smith et al., 2022). Climate change in northern Finland is expected to increase precipitation and temperature resulting in a near-complete loss (98.2%) of suitable conditions for palsas by 2080 (Borge et al., 2017; Fewster et al., 2022; Könönen et al., 2022). The loss of palsas will switch vegetation to deep-rooted plants such as sedges, change hydrology by increasing soil moisture (Hugelius et al., 2020; Malhotra and Roulet, 2015). Those changes modify C cycle dynamics, altering the C storage capacity of the ecosystem across the discontinuous permafrost zone (Hugelius et al., 2020; Kurylyk et al., 2016; Olefeldt et al., 2021).

During thaw processes, previously frozen soil rejoins the surrounding wetland. This newly thawed material becomes a deeper active layer and generally allows OM to be more readily accessible to microbial organisms (Schuur et al., 2008; Turetsky et al., 2002). This disturbance is known to result in increased mineralization of C as greenhouse gasses, such as carbon dioxide ($CO_2$) and methane ($CH_4$) (Kirkwood et al., 2021; Treat et al., 2014; Turetsky et al., 2002). Since it is usually almost entirely OM, C loss in permafrost peatland soils is also strongly affected by the degree of decomposition of the peat and the peat (Jones et al., 2017; Treat et al., 2014), making potential C loss estimates more complex.

The vertical dynamics of C decomposition in permafrost peatlands are especially not yet well understood. Earlier studies found net C accumulation following thaw in newly accumulated surface peat (Camill, 1999; Turetsky et al., 2007). However, more recent projections suggest that permafrost thaw could result in up to 30% soil C loss from the deeper, pre-thaw soil C stock (Hugelius et al., 2020; Jones et al., 2017). This would result in a net C loss for decades to centuries (Hugelius et al., 2020; Jones et al., 2017) or a balanced C system where increased productivity offsets decomposition (Heffernan et al., 2020). Incubation experiments simulated permafrost thaw to assess vertical OM decomposition over the thaw transect by measuring gas ($CO_2$ and $CH_4$) production at specific depths (Baysinger et al., 2025; Hodgkins et al., 2014; Kirkwood et al., 2019; Treat et al., 2014, 2015). They measured higher $CO_2$ production from the surface active layer than from the permafrost. Incubation studies generally agree that there is an increase in $CO_2$ production along the thaw stages (Baysinger et al., 2025; Hodgkins et al., 2014; Kirkwood et al., 2019; Treat et al., 2014, 2015). Additionally, permafrost layers contributed up to 40% of the total $CO_2$ production (including the active layer), even under anaerobic conditions. Field measurements, in contrast, indicate permafrost C contribution under dry conditions only, with most C contributions coming from the decomposition of modern C

(Estop-Aragonés et al., 2018a; Cooper et al., 2017). In addition, via new peat accumulation vegetation and water-logged soils in thermokarst fens offset C emissions from collapsing palsas, resulting in a C sink, contrasting with findings from more recent projections (Hugelius et al., 2020; Jones et al., 2017). Still, despite new peat accumulation, projections of peat C loss with warming and permafrost thaw range from 3.0 kg C m$^{-2}$ to 36 kg C m$^{-2}$ (Hugelius et al., 2020; Jones et al., 2017; Treat et al., 2021).

The discrepancies between laboratory incubations, on-site field measurements, and model projections highlight potential methodological limitations, particularly the need to study the whole peat profile. Indeed, incubations isolate layers of soil, often homogenizing them and thereby disturbing the soil matrix (Treat et al., 2015), which does not allow for accurate C emission quantification. While field conditions preserve the soil structure, simulating permafrost thaw under controlled conditions is nearly impossible. Field-based chronosequences of permafrost thaw can show the different thaw stages in permafrost peatlands that usually happen over several decades or centuries. However, chronosequence approach makes it difficult to study the transitions between thaw stages and to isolate specific processes. Mesocosm incubations combine controlled conditions with in-situ soil structure and could offer more realistic C loss estimates (Voigt et al., 2019). To our knowledge, only one recent study has simulated thaw in permafrost peatlands with mesocosm incubations. While the authors focused on vegetation and soil moisture, it emphasizes the need to better understand the links between C cycling and hydrology in permafrost peatlands (Voigt et al., 2019).

Thaw can proceed gradually over decades or abruptly within days to years, with the latter often forming thermokarst ponds at collapsing palsa edges (Quinton and Baltzer, 2013; Jorgenson et al., 2006; Borge et al., 2017). Abrupt thaw has recently been more precisely defined as occurring within 30 years in ice-rich soils (>20%) and/or when causing major ecological or state shifts such as wildfire or streamflow changes (Webb et al., 2025). This definition incorporates not only a temporal criterion but also environmental and soil factors such as hydrology and rapid soil moisture increase. With this updated definition, abrupt thaw appears to be the dominant form of permafrost degradation in palsas. In palsa mires, permafrost degradation leads to ground subsidence and hydrological changes, driving the transition from elevated palsas to wetter peatland ecosystems such as bogs and fens (Hugelius et al., 2020). Additionally, studies from Fennoscandia estimated palsa degradation rates between -1.0 % yr$^{-1}$ and -1.3 % yr$^{-1}$ over the period of 1950/60-2010/2014 (Borge et al., 2017; Leppiniemi et al., 2025; Olvmo et al., 2020) with palsa losses reaching up to 80 % between 2007 and 2021 in specific areas such as Finnish Lapland. Permafrost degradation was primarily attributed to palsa collapse rather than active layer deepening (Verdonen et al., 2023).

Although abrupt thaw affects a smaller area, it may contribute to permafrost C release- up to 40%— yet is underrepresented in Earth System Models, particularly regarding associated hydrological changes (Turetsky et al., 2020; Rodenhizer et al., 2023b). Most studies on abrupt vs. gradual thaw have focused on steady-state outcomes—dried peatlands or stabilized thermokarst features—measuring C release under either aerobic (gradual) or anaerobic (abrupt) conditions, with the latter typically showing lower $CO_2$ emissions (Baysinger et al., 2025; Harris et al., 2023; Hodgkins et al., 2014; Kirkwood et al., 2021; Jones et al., 2017). As a result, microbial, physical, and redox processes during the transitional thaw stage remain poorly understood. In transitional stages, soils are newly flooded yet not fully anoxic and this may also increase in-situ $CO_2$ emissions, as it has been measured in thermokarst (Kuhn et al., 2018; Matveev et al., 2016; McGuire et al., 2009; Rodenhizer et al., 2023b). Dur-

ing thaw transitions, interactions between collapsing palsas and adjacent fens create distinct redox conditions in thermokarst ponds, making the OM more available for microbial degradation (Patzner et al., 2020). Thermokarst ponds also support diverse microbial communities, favoring C mineralization (Crevecoeur et al., 2015; Leroy et al., 2025; Peura et al., 2020). In addition, flooding can enhance the vertical and lateral transport of labile carbon, temporarily increasing $CO_2$ emissions (Van Gestel et al., 1993; Kim et al., 2010). Most of these microbial, physical, and redox processes have been observed primarily under in-situ conditions, making it difficult to isolate specific drivers.

To understand how hydrological changes during abrupt and gradual permafrost thaw influence C emissions in palsa degradation, we incubated meter-long peat cores (mesocosms) from a Finnish palsa mire. Mesocosm incubations preserve in-situ soil horizon structure while maintaining a high-level of control over environmental conditions. With this setup, we simulated abrupt and gradual permafrost thaw under different hydrological conditions using thermokarst pond water to assess microbial and redox influences on C release. We continuously monitored $CO_2$ and $CH_4$ emissions and measured peat decomposition using Fourier Transform Infrared (FTIR) spectroscopy. In addition, we incubated mesocosms from a post-thaw fen to assess the long-term vs. short-term thaw effects on C releases.

Based on this design, we hypothesized that (H1) flooding with abrupt thaw alters vertical C dynamics by physically disturbing the soil matrix and changing the redox properties of the soil to anaerobic conditions. Anaerobic conditions under the abrupt thaw simulations result to lower $CO_2$ emissions compared to gradual thaw. (H2) The addition of thermokarst pond water during the thaw simulation enhances C emissions through increase microbial activity due to microbial colonization from the thermokarst pond water. (H3) Long-term thaw in the fen results in higher C emissions than in the palsa because of fresh organic matter input, permafrost-free conditions for several decades and therefore already established microbial communities. The in-situ water logged conditions in the fen site allow for $CH_4$ emissions during the incubation, while under abrupt thaw simulation, ideal redox conditions are not reached after three-month incubation despite water saturation and therefore no $CH_4$ emission occurs.

## 2   Methods

### 2.1   Study site

We collected cores for the mesocosm incubations from a palsa mire located in a discontinuous permafrost area in the Finnish part of Sápmi in March 2023. The Peera palsa mire (68.8778°N; 21.0792°E) is composed of four palsas (Verdonen et al., 2023), surrounded by fens. The mean annual air temperature measured at the Kilpisjärvi station is -1.7 °C, and the average annual precipitation (rain and snow) is 546 mm (1991–2020, FMI, 2024). In this study, we focused on the southernmost landform (Fig. 1). Over more than 60 years of aerial imagery monitoring, the palsa has lost 77% of its surface area, transitioning into thermokarst bog and fen habitats. This transition is associated with a shift in vegetation from shrub-dominated peat to open water, followed by colonization by *Sphagnum* mosses and sedges (Verdonen et al., 2023). Based on the definitions from Peura et al. (2020), the palsa is surrounded by three developing thermokarst ponds and four emerging ponds at its edge. The thermokarst ponds are located at the same elevation as the surrounding fen, which lies approximately two meters lower than the

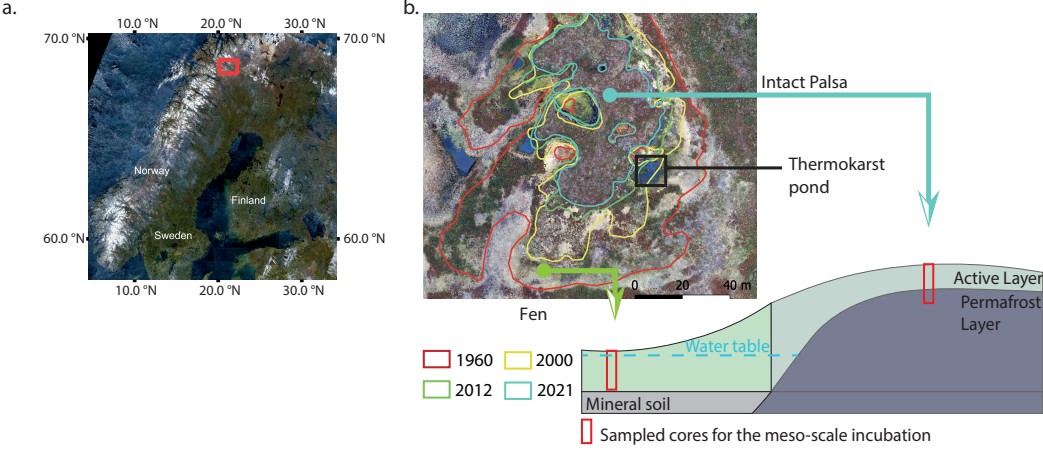

**Figure 1.** Location of the Peera palsa study site. Aerial image of Finland (Sentinel-2 Mosaics, Copernicus) with the site marked by a red square (a); Aerial view of the palsa (Verdonen et al., 2023) and schematic of the topography and sampling locations for the mesocosm incubation. Lines on the aerial view show the extent of the palsa at different years (b).

palsa. We measured a peat thickness of 1.7 m and an active layer depth of 60 cm at the top of the Peera palsa dome. The active

layer was measured during the first exploratory expedition in October 2022. Since this was at the end of the growing season, we assume this represents the maximum annual active layer depth. The vegetation at the sampling locations on Peera palsa is characterized by shallow-rooted species such as *Betula nana subsp.*, *Empetrum nigrum subsp.*, and *Vaccinium myrtillus*, while the surrounding fen is dominated by deep-rooted vegetation such as *Eriophorum*, *Sphagnum*, and sedges (Fig. S1).

### 2.2 Sampling of peat mesocosms and water

We sampled cores from two thaw stages: the intact palsa and the thermokarst fen (Fig. 1). To minimize soil disturbance, sampling was conducted in winter (March 2023), when frozen conditions help preserve water content and prevent artificial compaction. We collected twelve cores (1 m long, 7.5 cm diameter) from the Peera palsa and three from the fen (Fig. 1). At both sites, we also collected a 'long core' down to mineral soil for sedimentary analysis. Mesocosms from the palsa were drilled using a snow, ice, and permafrost (SIPRE) auger (Jon's Machine Shop, Alaska, USA). In the fen, only the top 15 cm

were frozen, so we used the SIPRE auger for that layer and then drove a metal pipe into the unfrozen soil to complete the core. To preserve soil structure, the pipes were frozen vertically. We wrapped the palsa samples and the frozen top layers from the fen in core foil and stored them in thermoboxes.

We collected 45 L of water from a thermokarst pond adjacent to the palsa (68.8775°N; 21.0798°E), using pre-cleaned 10 L canisters. We drilled through the surface ice with the SIPRE auger before sampling.

Samples were stored at -20°C at the Kilpisjärvi Research Station during the expedition. They were later shipped frozen to AWI in Potsdam, Germany, using a refrigerated truck and stored at -20 °C until the start of the experiment.

### 2.3 Mesocosm set-up

### 2.3.1 General setup

This study aimed to quantify $CO_2$ and $CH_4$ emissions under both abrupt and gradual permafrost thaw simulations, focusing
on hydrological changes during abrupt thaw. We used a mesoscale incubation setup to preserve the in-situ soil structures and
allow gas diffusivity in the soil column. We incubated meter-long cores (mesocosms) from the two sampled sites, the palsa and
the fen, at 10°C under dark conditions for 12 weeks (Fig. 2). We sequentially thawed permafrost to simulate palsa degradation
and applied the same treatment to the fen cores. For all the mesocosms, $CO_2$ and $CH_4$ emissions were continuously measured
throughout the incubation duration (2.4). The mesocosms from the palsa were exposed to four different treatments simulating
gradual and abrupt thaw (see details below). A core from the fen was included as an end-member representing long-term
thawed conditions ( 60 years). This allowed comparison of $CO_2$ and $CH_4$ emissions following short-term simulated permafrost
thaw with those from a naturally thawed fen. In the field, the water table at the fen site was at the surface (September 2022).
Therefore, to best represent in-situ conditions, we kept the water table of the mesocosms from the fen at the surface by adding
water from the thermokarst pond. For all the mesocosms, we removed the vegetation at the surface of the core. We used three
replicates for each treatment, for a total of fifteen mesocosms.

### 2.3.2 Gradual vs. Abrupt thaw treatments

We simulated gradual thaw (control treatment; Fig. 2) by incubating mesocosms from the palsa site under in-situ moisture,
which was approximately 60% volumetric water content (Fig. S13). When palsas thaw abruptly, soil typically collapses into
thermokarst ponds (Fig. 2). To simulate the sudden increase in soil moisture during abrupt thaw, we flooded the mesocosms
until the water table reached the surface (0 cm). We chose to flood the mesocosm with a similar water table as the one measured
at the fen site to be able to compare emissions from both sites. For both thaw simulations (gradual and abrupt thaw) mesocosms
were thawed at the same speed and only the water content was modified. The mesocosms were subjected to three treatments to
test the influence of water chemistry and microbes: autoclaved tap water (TW), non-filtered (NF) thermokarst pond water, and
filtered (F) thermokarst pond water. The NF treatment exposed the mesocosms to both the microbial and chemical properties
of the thermokarst pond water. The F treatment, in which microbes were removed by filtration (SI) from the thermokarst pond
water, isolated the chemical properties of the thermokarst pond water. To isolate the effect of flooding on C emissions, we
flooded another set of mesocosms with the TW treatment.

### 2.3.3 Water addition

For all mesocosms under flooded conditions (TW, NF, F, and the fen), we flooded the mesocosms from the side at four different
depths (Fig. S5). We added water from the side to avoid preferential vertical pathways. We used Bev-A-Line tubes (Saint-
Gobain 06490-12) equipped with three-way valves (Masterflex®-Fitting, MFLX30600-25). We thawed the samples overnight,

and added water the day after. After each thaw stages, we applied the same procedure. Additionally, we ensured the water table remained at the surface by bi-weekly checks and adjusted the level as needed (Fig. S6).

### 2.3.4 Sequential thawing

All cores were thawed in three steps to assess the contribution of C emissions from the active layer (60 cm thick), which thaws during the growing season, and the permafrost layers. To isolate the effects of hydrological changes between gradual and abrupt thaw, all cores underwent the same permafrost thawing process. The sequential thaw was implemented by exposing the top 60 cm to 10°C for four weeks, ensuring complete thaw of the active layer. The first thaw stage corresponded to the maximal annual active layer depth measured in October 2022, and the incubation temperature (10°C) corresponded to the mean growing season

soil temperature. To thaw the permafrost, we deepened the active layer by 20 cm every four weeks (Table S2). The duration of each sequential thaw was determined based on the time need for $CO_2$ fluxes to stabilize during the first thaw thaw step. For consistency, we applied the same thaw duration for each thaw step. However, we acknowledge that the duration for $CO_2$ fluxes to stabilize following thaw might have differed for each sequential thaw. The frozen part of the core was kept at -5°C. Due to setup limitations, we also applied the sequential thawing to the mesocosms from the fen, although this did not fully represent

field conditions. Quality control of conditions was performed by continuously monitoring the temperature inside the incubator and the soil for one replicate per treatment. The temperature was recorded at five depths (0–20–40–60–80 cm) every half hour using temperature sensors (Hobo U12-008 data logger, Fig. S7).

The details regarding the mesocosms, water preparation and addition, and incubation system can be found in the Supple-

185 mentary Information (Fig. S3 and Fig. S4).

## 2.4 $CH_4$ and $CO_2$ fluxes

### 2.4.1 Flow-through steady-state system

We used a flow-through steady-state system to measure the concentrations of $CH_4$ and $CO_2$ in the mesocosm headspace. As a reference for flux calculations (Eq. 1), an additional empty incubation vial was included and flushed with ambient air. Each

190 channel was connected to a Picarro G2508 gas analyzer (PICARRO, INC., Santa Clara, USA) and measurements were taken over an eight-minute period. When a mesocosm was being measured, a subsampler redirected air from its headspace to the gas analyzer for concentration analysis. The measured GHG concentrations represent the increased GHG levels in the mesocosm headspace, which reflect both the equilibrium of GHGs emitted from the mesocosm and the replacement of headspace air. At the end of each measurement cycle, the system switched to the next mesocosm, including the reference channel. When

not being measured, the mesocosms' headspaces were continuously flushed with a stable flow of ambient air (flow rate = 16 mL.min$^{-1}$). To prevent desiccation of the mesocosms, the inflow air was saturated with water (Fig. S4).

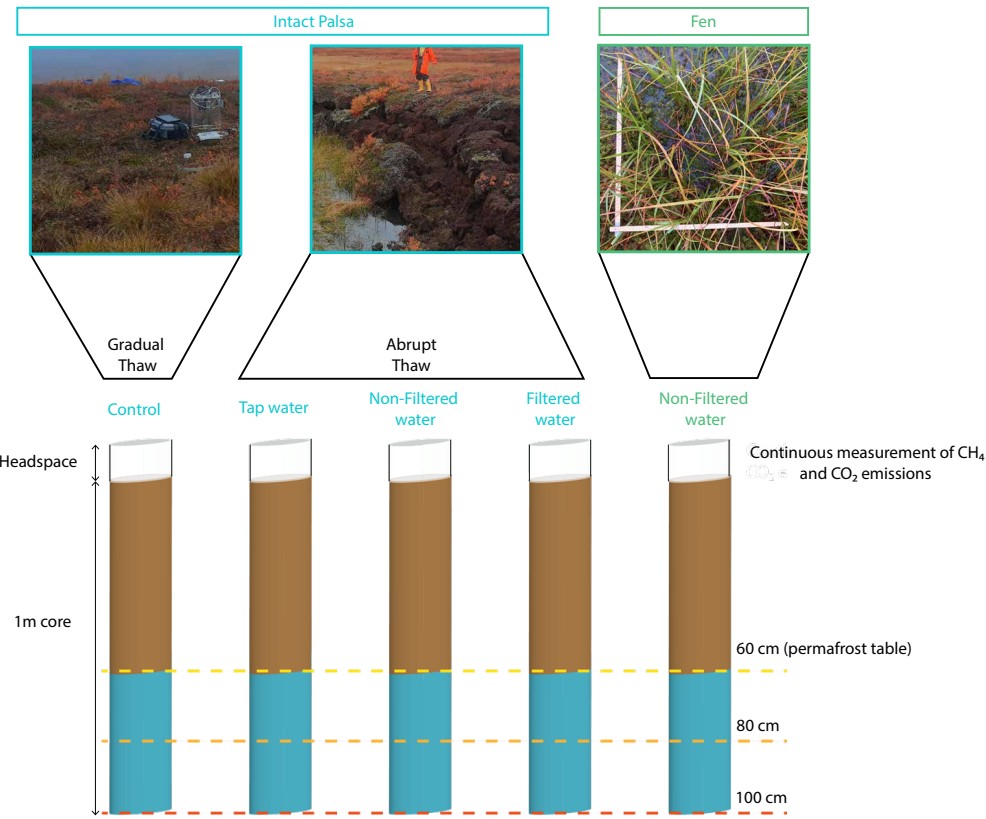

**Figure 2.** Schematic of the mesocosm setup used for the mesocosm incubation of the samples from Peera, Lapland. The brown part of the cores was directly thawed at the beginning of the incubation, while the blue part was sequentially thawed. For the intact palsa, the blue part corresponds to the permafrost layer. The dashed lines indicate the three thaw stages during the incubation period.

### 2.4.2 Quality Control prior to Flux calculations

During incubation, we measured $CO_2$ fluxes above the gas analyzer's detection limit (5,000 ppm). Fluxes above this threshold saturated the C sensors, rendering the system unable to measure $CH_4$ emissions. We manually filtered out time windows for each channel where flux anomalies were detected (Supplementary Info, Fig. S5, Table S1). Overall, 11.7% of the data were filtered (Table S1, Fig S18 and S19).

### 2.4.3 Flux calculations

Data were processed prior to flux calculations. We excluded the first five minutes of each measurement to ensure that we were capturing emissions from the mesocosms, rather than carryover air from the system. C fluxes were calculated as the difference between the C emissions from the mesocosm headspace (outlet) and the reference channel (inlet) over time. The concentrations

were converted using the Ideal Gas Law and then multiplied by the flow rate to obtain the fluxes (Mastepanov and Christensen, 2009).

$$F = \frac{\Delta C f p M}{S R T} \tag{1}$$

Where $F$ the flux (mg m$^{-2}$ h$^{-1}$), $\Delta C$ is the difference gas concentration between the measured mesocosm and the reference channel in ppm, $f$ the flow rate (m$^3$ h$^{-1}$), $p$ the pressure (Pa), $M$ the molar mass (in mol g$^{-1}$), $S$ the base area of the mesocosm (in m$^{-2}$), $R$ the ideal gas constant (m$^3$ Pa K$^{-1}$ mol$^{-1}$) and $T$ the incubation temperature (K). We adapted the script from Hoffmann et al. (2017) to calculate the C fluxes.

### 2.4.4 Quality control post-flux calculation

We identified outliers by applying a rolling Interquartile Range (IQR) with a sliding step of one day. Outlier measurements resulting from laboratory manipulation or technical issues were excluded. After data cleanup, we quantified fluxes below the instrument's detection limit by calculating the Minimum Detectable Flux (MDF) (Nickerson, 2016). The final dataset is available in Laurent et al. (2023b).

## 2.5 Soil analyses

### 2.5.1 Sedimentary and pore water analysis

We characterized the sedimentary and geochemical composition (including TOC, TN, conductivity, pH, DOC and water content) of the intact palsa and fen sites using long cores. We thawed the samples at room temperature and extracted the pore water using a rhizon soil moisture sampler (Rhizon MOM 0.6 $\mu$m, Rhizosphere Research Product – Meijboom and van Noordwijk, 1991) for two days. We separated the water collected into two 50 mL bottles, one for pH and Electrical Conductivity and the second one for Dissolved Organic C (DOC). EC and pH measurements were conducted by a conductivity pocket meter with a reference temperature of 25 °C (Cond 340i, WTW, Germany) and a potentiometer (Multilab 540, WTW, 125 Germany). The water for DOC was acidified with 30 percent HCl and was analyzed by catalytic combustion at 630 °C (TOC-VCPH,Shimadzu, Japan).

We measured Total Organic Carbon (TOC), Total Carbon (TC), and Total Nitrogen (TN) before and after incubation at five different depths along the core (0 cm, 25 cm, 50 cm, 75 cm, and 100 cm). We used an elemental combustion analyzer (Elementar soliTOC cube) to measure TOC and TC, and a nitrogen analyzer (Elementar rapidMAX N, Germany) to measure TN. We analyzed each subsample in duplicate to ensure precision and used standards and blanks to maintain reliable and accurate analytical measurements. The final dataset reporting the geosedimentology of the samples can be found in Laurent et al. (2023a).

### 2.5.2 Water content

We measured the initial gravimetric soil water content using the samples collected for sedimentary analysis. We first weighed the samples while they were still frozen, then weighed them again after freeze-drying. We calculated gravimetric soil water content as the mass of water lost divided by the mass of the dried soil.

### 2.5.3 Fourier Transform Infra Red analysis

We analyzed the same samples as for TOC and TN analyses for FTIR spectroscopy in the KBr technique (Heller et al., 2015; Ellerbrock et al., 2024). The samples were measured at the Leibniz-Zentrum für Agrarlandschaftsforschung, Müncheberg (Germany). We used the average of 16 scans to obtain the spectra (Ellerbrock et al., 2024). Each sample was corrected against ambient air (Haberhauer and Gerzabek, 1999). The baseline corrected spectra (Ellerbrock et al., 1999) were interpreted as described in Ellerbrock and Gerke (2013).

To characterize the peat decomposition, the interpretation of the FTIR spectra focused on absorption bands that reflect the amount of aliphatic (CH) and carboxylic (C=O) groups within the peat samples. The CH band (2900-2700 cm$^{-1}$) is characteristic of the OM aliphatic backbone (Ellerbrock and Gerke, 2013). While the C=O band (1710-1640 cm$^{-1}$) reflects the presence of oxidized (carboxyl) groups. A higher intensity in the C=O band signifies increased OM decomposition. CH bands are usually superimposed as a shoulder of the broad OH band which spans 3750-2700 cm$^{-1}$. The absorption intensities of the CH bands were quantified as the vertical distance from a local baseline (Capriel et al., 1995), which was defined between tangential points on the OH-band (near the wave numbers 3020 and 2800 cm$^{-1}$ (red arrows within the enclosure in Fig. S11; Ellerbrock et al. (2009)). Similarly, the C=O band intensities (black arrows in Fig. S11) were measured as the height from the total baseline to the maxima within wave number ranges 1780–1710 cm$^{-1}$ and 1680–1580 cm$^{-1}$ (Ellerbrock and Kaiser, 2005).

The ratios of specific bands provide insights into OM characteristics. The ratio of C=O to polysaccharide (COC) (C=O/COC ratio) indicates the potential cation sorption capacity (Ellerbrock and Gerke, 2021), while the ratio of CH to COC (CH/COC ratio) reflects the balance between non-oxidized OM and litter OM, serving as a proxy for potential hydrophobicity. To evaluate the decomposition stage of the peat samples caused by incubation process we adopted here a procedure described by Hodgkins et al. (2014) by relating the CH/COC and C=O/COC ratios of the original peat samples with those of the incubated samples (Eq. 2, Eq. 3). One replicate among the three was used for the post-FTIR analysis. The replicate with temperature sensors could not be used due to potential contamination because of the sensors. The third one was kept intact as a backup.

$$DI_{CH} = \frac{CH/COC_{\text{after incubation}}}{CH/COC_{\text{before incubation}}} \tag{2}$$

$$\text{DI}_{C=O} = \frac{\text{C=O/COC}_{\text{after incubation}}}{\text{C=O/COC}_{\text{before incubation}}} \qquad (3)$$

With $\text{DI}_{CH}$ representing the decomposition indices (DIs) based on changes in the aliphatic backbones, and $\text{DI}_{C=O}$ assessing
changes in the carboxylic group content. An index >1 means an increase of the decomposition during the incubation.

## 2.6 Data processing and statistical methods

We characterized the thawing response by determining shifts in the emissions over time. To do so, we applied a changing-
point-based method coupled to a sliding mean (Rodionov, 2004). When a potential shift is detected, a Student's t-test is run
to validate whether the anomaly is statically different from the mean value. We tested the method for several cut-off lengths
to cover short-term and long-term regimes. In the end we chose a cut-off length of 5 days. We ran the Rodionov test with the
rshift package (Room et al., 2020).
We tested the differences among treatments and thaw stages for the averaged fluxes with a one-way ANOVA coupled to a
TukeyHSD test. We also grouped the treatments between flooded and dry conditions and tested for significance. Before per-
forming statistical test, we tested the data for independency, normality and homogeneity of variance. Finally, to ensure that
filtering did not affect the statistics, we performed a Kruskal-Wallis test followed by pairwise comparisons using Dunn's test
on both the non-filtered (before quality control) and filtered (after quality control) datasets. Because extreme values can strongly
influence mean-based comparisons, we employed robust, median-based statistical methods that are less sensitive to outliers to
compare the two datasets (Table S4).

We used R (R Core Team, 2024) for all the computations.

## 3 Results

### 3.1 Soil and Pore Water Characteristics

pH values in the active layer ranged from 3.4 to 3.6. The pH in the permafrost layer was less acidic, with values between 4.2
(depth 100 cm) and 4.9 (permafrost table). EC was three times higher in the active layer than in the permafrost layer. As with
pH and EC, the DOC concentration increased in the permafrost layer (ranging from 120 to 165 mg·L$^{-1}$ in the active layer and
from 402 to 480 mg·L$^{-1}$ in the permafrost layer) (Fig. 3). The pH and EC in the fen core were homogeneous along the soil
profile, with pH values similar to those in the permafrost layer and EC values of the same order of magnitude as those in the
active layer. Unlike in the palsa, DOC concentration in the fen decreased with depth (Fig. 3). TOC and C:N ratios did not differ
between the two sites. TOC values ranged from 41% to 52%, with the lowest values (41%) measured at the permafrost table.
The C:N ratios remained consistent along the soil profile (between 30 and 40), with a decrease at the permafrost table and at a
depth of 70 cm (16) (Fig. 3).

Overall, we observed a difference between the active and permafrost layers in the soil pore water, while the sediment parameters remained stable along the soil profile. The soil parameters in the fen were mostly stable with depth. Finally, the difference between the palsa and the fen was mainly evident in the composition of the soil pore water.

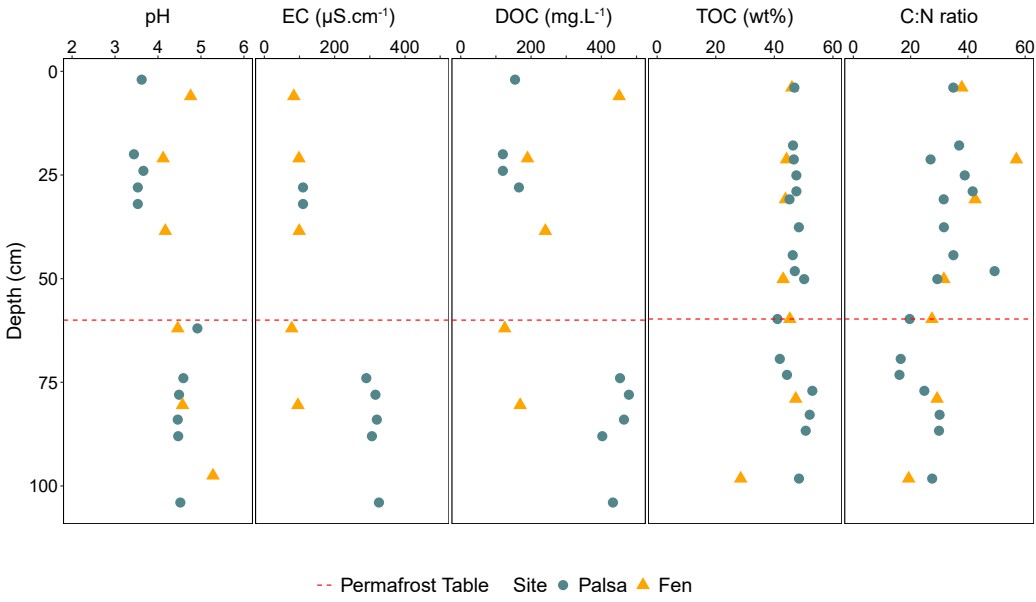

**Figure 3.** Sediment and soil pore water parameters within the first meter of the palsa and fen prior to incubation. Green circles indicate values measured from the palsa, while yellow triangles represent values from the fen. The red dashed line marks the position of the permafrost table in the palsa.

## 3.2 FTIR - Organic matter quality and decomposition

The FTIR spectra of the palsa samples collected at 0–2 cm and 18–22 cm depth showed band intensities 15% (0–2 cm) to 30% (18–22 cm) higher for the CH and C=O bands (Fig. 4a – 0 and 20 cm), compared to the fen samples. At 60 cm depth (the permafrost table), the opposite trend was observed (Fig. 4a – 60). No differences were observed in the FTIR spectra of palsa and fen samples collected at 100 cm (Fig. S12). After incubation, the strongest decomposition of OM was observed in samples from the permafrost table (60 cm), with a DI of 1.8 (Fig. 4b). This was also supported by an increase in the band intensities of CH and C=O groups post-incubation (Fig. 4b). The DI ratios in the flooded (abrupt thaw simulation) cores were approximately 1.7 times higher than those in the Control (Fig. 4b; Fig. S12). Little changes were observed in other layers. In the fen cores, the most significant increase in band intensity post-incubation was detected in the bottom layer (100 cm), which also had the highest DI value of 1.9. Above this depth (0–80 cm), little to no change was observed during incubation.

Under in-situ conditions (i.e., prior to incubation), we measured a higher degree of OM decomposition in the active layer of the palsa compared to the fen. After incubation, the depth of greatest OM decomposition varied by site: in the palsa, the permafrost

table experienced the highest decomposition, whereas in the fen, most changes in CH and C=O bands occurred in the bottom layer.

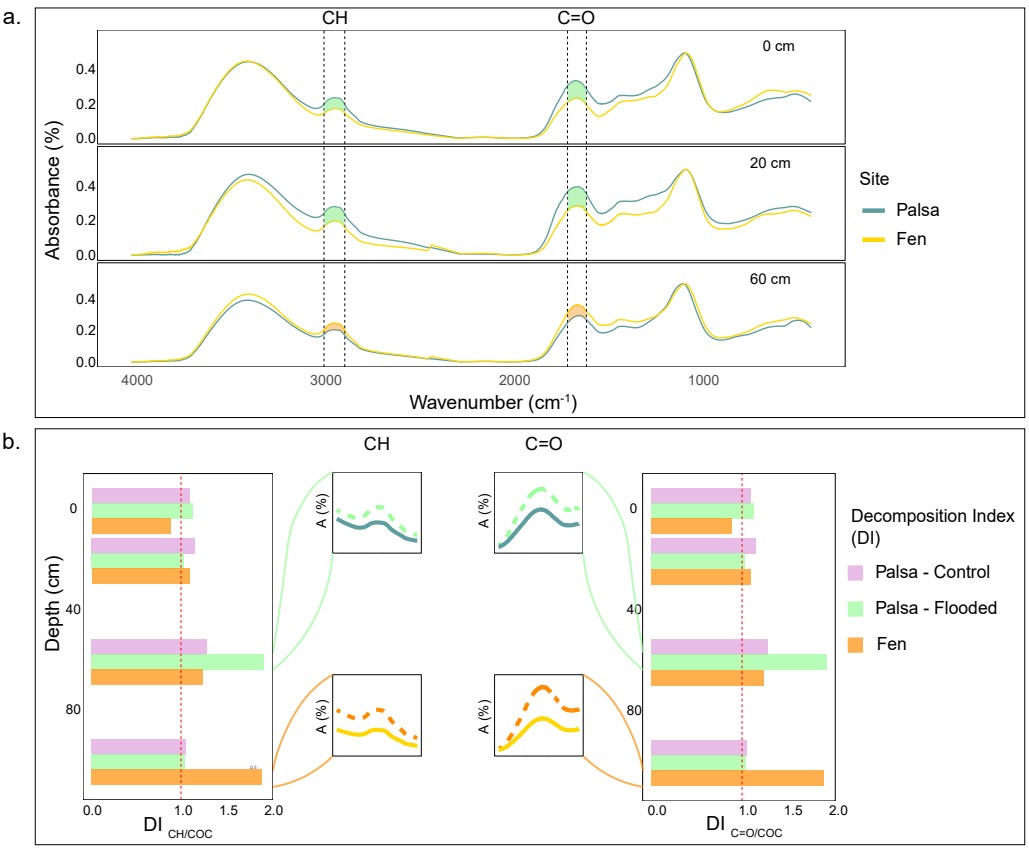

**Figure 4.** Peat quality along the the soil profile of the palsa and the fen. FTIR spectra of peat samples collected at the palsa and fen sites from 0 to 2 cm (top), 18 to 22 cm (center), and 58 to 62 cm (bottom) (a.); Decomposition Index (DI) of CH/COC (left) and the C=O/COC ratios (right) during incubation were calculated from FTIR data for palsa and fen samples, both before and after incubation (Hodgkins et al., 2014). The zoomed-in spectral sections highlight differences in band intensity before (solid line) and after (dashed line) incubation, focusing on the depth with the greatest increase in degradation (b).

## 3.3 CO$_2$ emission dynamics

### 3.3.1 Transient palsa thaw vs. stabilized thaw fen incubated under in-situ conditions

During the incubation period, CO$_2$ emissions in the fen mesocosms were consistently three to four times higher than in the Control (Fig. 5, Table 1). The Control reached its highest emissions (233 mg-CO$_2$·m$^{-2}$·h$^{-1}$) at the end of the incubation (86 days), whereas the fen attained its maximum (587 mg-CO$_2$·m$^{-2}$·h$^{-1}$) after 37 days (Fig. 5).

We observed a CO$_2$ burst from the fen mesocosms during the first seven days, followed by a continuous decrease (Fig. 5,

Fig. S14). For the second and third thawing events, a transition period—defined as the period following a thaw event when C emissions decrease or increase before stabilizing—lasted three and two days, respectively. This was marked by a 66% decrease in $CO_2$ emission rates. Following these periods, $CO_2$ emissions increased again and reached a new steady state (Fig. 5). In contrast, in the Control, deepening of the active layer triggered a four-day $CO_2$ burst during the second thaw event (120 mg-$CO_2 \cdot m^{-2} \cdot h^{-1}$). Thereafter, emissions returned to a stable rate ( 15 mg-$CO_2 \cdot m^{-2} \cdot h^{-1}$). The transient period during the third thaw event was similar to the fen cores (two days); however, unlike the second thaw event, $CO_2$ emissions increased continuously after the transient period, rising from 51 to 233 mg-$CO_2 \cdot m^{-2} \cdot h^{-1}$.

The average $CO_2$ emissions from the Control showed no significant differences between thaw stages (TukeyHSD; p-value > 0.05). However, the last event exhibited a threefold increase in $CO_2$ emissions compared to the second (0–80), due to deepening of the permafrost table (Table 1). When comparing the normalized volume of thawed soil, $CO_2$ emissions were of the same order of magnitude for the first and last thaw stages (0–60: 94.3 $\pm$ 26.3 mg-$CO_2 \cdot m^{-3} \cdot h^{-1}$; 0–100: 98.3 $\pm$ 3.6 mg-$CO_2 \cdot m^{-3} \cdot h^{-1}$; Table 2). For the fen, we measured a continuous 85% increase in average $CO_2$ emissions with thaw stage (Table 1) and stable $CO_2$ emissions per volume of thawed soil (Table 2)

Overall, in mesocosms incubated under in-situ conditions, we observed two distinct behaviors based on site. $CO_2$ emissions from the Control were significantly lower than those from the fen (TukeyHSD; p-value = 0.0004; Table 1). Finally, we measured a stronger response to thaw deepening in the fen compared to the Control.

### 3.3.2  Palsa degradation under different hydrological conditions

For the three thaw stage periods, we did not observe significant differences in average $CO_2$ emissions between the water treatments (TukeyHSD; p-value > 0.05; Table 1, Fig. 6). However, we did observe significant differences when we grouped all the water treatments together, compared to the Control (TukeyHSD; p-value = 0.005; Table 1, Fig. 6). Throughout the incubation, the mesocosms under flooded conditions had emissions two to three times higher than those under control conditions, except for the Tap water during the second thaw stage (0–80). Emission rates ranged between 120 $\pm$ 20.8 mg-$CO_2 \cdot m^{-2} \cdot h^{-1}$ (Filtered - 0–80) and 191.7 $\pm$ 57 mg-$CO_2 \cdot m^{-2} \cdot h^{-1}$ (Filtered - 0–60) (Fig. 6, Table 1).

After thawing the active layer, the palsa mesocosms simulating the abrupt thaw (under flooded conditions: Tap water, Non-filtered, Filtered Water) experienced a burst of $CO_2$ emissions (> 300 mg-$CO_2 \cdot m^{-2} \cdot h^{-1}$ for all three treatments) during the first two to three days (Fig. 5, SI). Afterward, the mesocosms reached stable emissions, ranging from 120 mg-$CO_2 \cdot m^{-2} \cdot h^{-1}$ (Tap water) to 160 mg-$CO_2 \cdot m^{-2} \cdot h^{-1}$ (Filtered water). These emission rates were within the range of $CO_2$ emissions observed in fen cores during the initial thawing period (Table 1, Fig. 6).

Thawing the permafrost did not result in increased $CO_2$ emissions for any of the cores under flooded conditions (Table 1, Fig. 6). When normalized by the volume of thawed soil, we measured significantly higher $CO_2$ emissions during the first thaw stage (0–60) under the abrupt thaw simulation (TukeyHSD; p-value = 0.02; Table 2). For individual water treatments, the highest $CO_2$ emissions were mainly observed during the first thawing step (0–60 cm), with maximum values for Tap water: 855 mg-$CO_2 \cdot m^{-2} \cdot h^{-1}$ (1-day); Non-filtered: 591 mg-$CO_2 \cdot m^{-2} \cdot h^{-1}$ (1-day); Filtered: 828 mg-$CO_2 \cdot m^{-2} \cdot h^{-1}$ (54-day). In contrast, the distribution of $CO_2$ emissions for the Control showed higher rates when the entire soil column was thawed (Table 1, Fig.

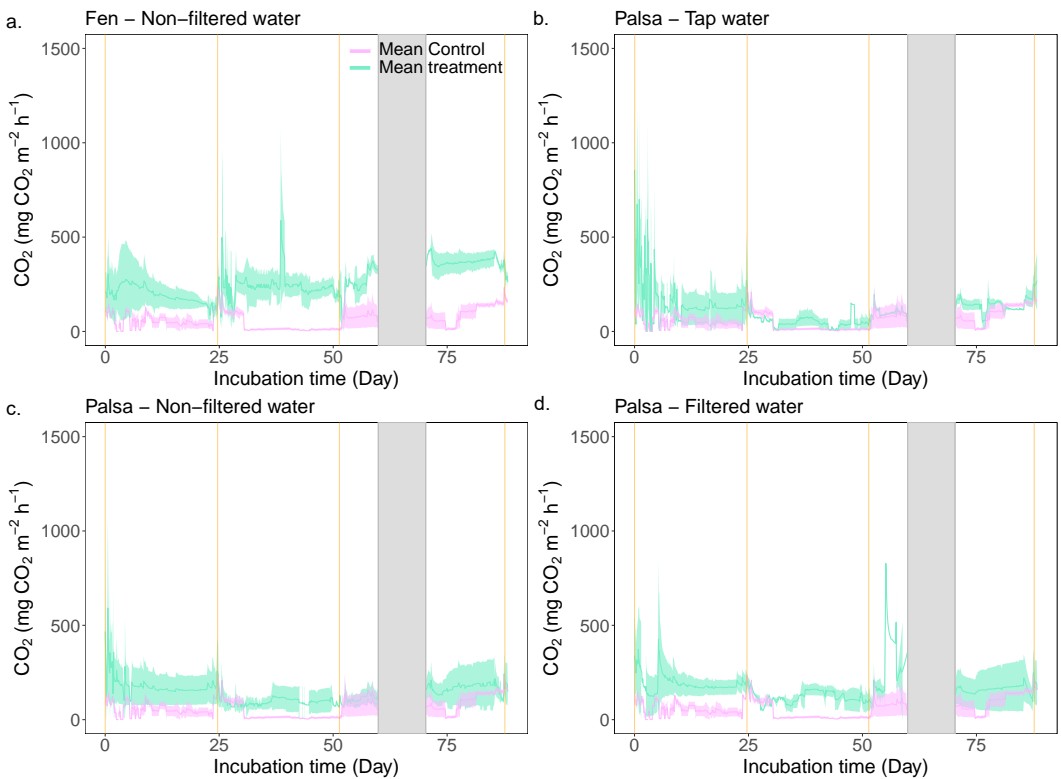

**Figure 5.** Continuous $CO_2$ emissions throughout the incubation period. The green line is the average $CO_2$ emission from the three replicates for the labeled treatment. The purple line depicts the average $CO_2$ emissions from the Control treatment and is used as a baseline. Finally, the vertical orange lines indicate the thaw stage times and the shaded grey area corresponds to a system failure.

6). Interestingly, we measured the minimum average $CO_2$ emissions during the second thawing event (0–80 cm) for all the
350 Palsa cores (Control and Flooded; Table 1).

Similar to the fen, we observed a 60 to 80% decrease in $CO_2$ emissions following the second thaw stage (0–80) for all mesocosms under flooded conditions. Two to three days later, $CO_2$ emissions increased again and eventually reached a new equilibrium (Fig. 5).

In summary, the flooded palsa cores exhibited $CO_2$ emissions that were twice as high as the Control cores. However, we did
not detect an effect of the different water treatments on the $CO_2$ emissions. The mesocosms under flooded conditions were not affected by thaw deepening, while the Control showed higher $CO_2$ emissions when the entire column was thawed.

**Table 1.** Average $CO_2$ and $CH_4$ emissions for each treatment and site over individual thaw stage. Averages not sharing any letters are significantly different by the Tukey-test at the 5% level of significance.

| Site | Treatment | Average $CO_2$ emissions (n=3) (mg-$CO_2$ m$^{-2}$ h$^{-1}$) | | | Average $CH_4$ emissions (n=3) ($\mu$g-$CH_4$ m$^{-2}$ h$^{-1}$) | | |
| --- | --- | --- | --- | --- | --- | --- | --- |
| | | \multicolumn Thaw Stage (cm) | | | | | |
| | | 0-60 | 0-80 | 0-100 | 0-60 | 0-80 | 0-100 |
| Palsa | Control | $56.6\pm15.8^a$ | $32.8\pm0.8^a$ | $98.3\pm3.6^{ab}$ | $-10.8\pm4.2$ | $-3.2\pm0.6$ | $-15.5\pm2.3$ |
| | Tap Water | $155.8\pm96.6^{ab}$ | $52.9\pm17.8^a$ | $144.1\pm17.9^{ab}$ | $-25.6\pm26.8$ | $-7.3\pm7.2$ | $-20.9\pm6.3$ |
| | Non-Filtered Water | $178.8\pm96.1^{ab}$ | $96.1\pm48.0^{ab}$ | $171.1\pm90.4^{ab}$ | $-18.1\pm31.8$ | $-27.9\pm18.5$ | $-34.6\pm15.4$ |
| | Filtered Water | $191.7\pm57.9^{ab}$ | $120.8\pm20.8^{ab}$ | $163.5\pm102.4^{ab}$ | $-33.0\pm21.4$ | $-31.1\pm17.1$ | $-25.1\pm21.2$ |
| Fen | Non-Filtered Water | $194.8\pm74.9^{ab}$ | $240.5\pm50.4^{bc}$ | $368.1\pm39.6^c$ | $9.7\pm17.8$ | $-33.5\pm2.9$ | $-53.9\pm4.1$ |

**Table 2.** Average $CO_2$ and $CH_4$ emissions for each treatment and site over individual thaw stage normalized by volume of thawed soil. Averages not sharing any letters are significantly different by the Tukey-test at the 5% level of significance.

| Site | Thaw simulation | Average $CO_2$ emissions (n=3) (mg-$CO_2$ m$^{-3}$ h$^{-1}$) Thaw Stage (cm) | | |
| --- | --- | --- | --- | --- |
| | | 0-60 | 0-80 | 0-100 |
| Palsa | Control | $94.3 \pm 26.3^{ab}$ | $40.9 \pm 0.9^{a}$ | $98.3 \pm 3.6^{ab}$ |
| | Flooded | $292.4 \pm 126.2^{d}$ | $112.4 \pm 50.8^{ab}$ | $159.6 \pm 69.9^{bc}$ |
| Fen | None | $324.6 \pm 124.8^{cd}$ | $300.6 \pm 63.0^{cd}$ | $368.1 \pm 39.4^{d}$ |

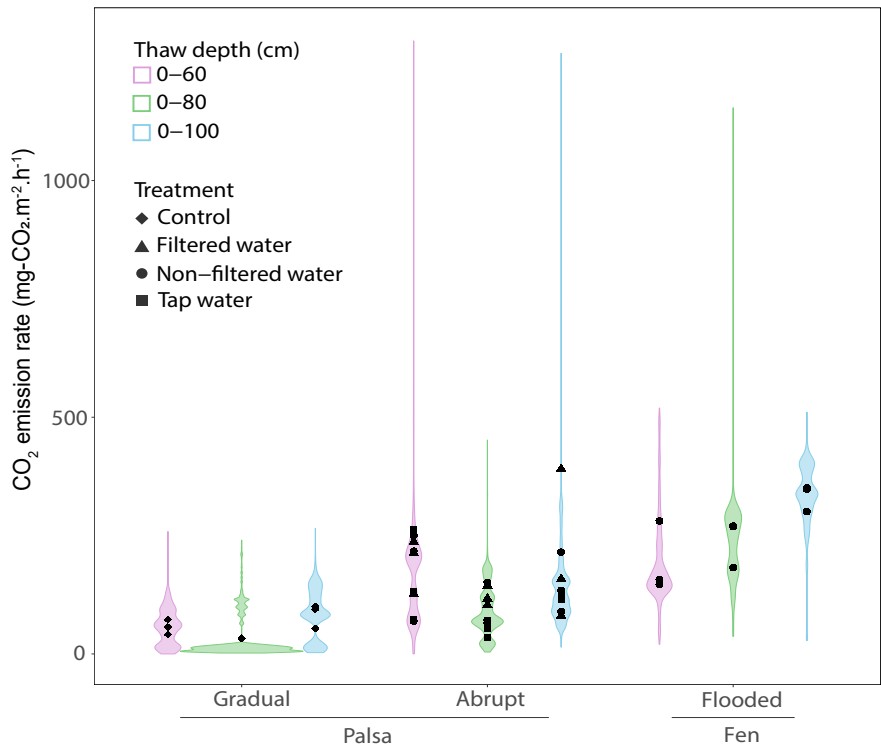

**Figure 6.** $CO_2$ emissions rates for each thaw stages under flooded or controlled conditions. The distribution uses the time series data of $CO_2$ emissions from each thaw stage, under conditions (control vs. flooded) and across sites (palsa vs. fen) to create the violin plots with the points indicating the average $CO_2$ emissions for each treatment. The colors show the different thaw stage events.

### 3.4 $CH_4$ emission dynamics

The palsa mesocosms exhibited a net uptake of $CH_4$ for all treatments (ranging from $-3.2 \pm 0.6$ to $-34.6 \pm 15.4$; respectively Control-0–80 and Non-Filtered 0–100). Neither the thaw stages nor the treatments had a discernible effect on the $CH_4$ uptake
rates (TukeyHSD, p-value > 0.05; Table 1).

For the fen, we observed a burst of $CH_4$ emissions during the first week of incubation, followed by emissions close to zero (Fig. S15). This resulted in an average $CH_4$ emission of $9.7 \pm 17.8$ $\mu$g-$CH_4 \cdot m^{-2} \cdot h^{-1}$ during the first thaw stage (0–60). The second thaw stage (0–80) triggered another burst of $CH_4$ emissions for three days, while no change in $CH_4$ emissions was observed after the third stage (Fig. S15). In the fen, the average $CH_4$ emissions for each thaw stage indicated a continuous increase in
$CH_4$ uptake throughout the incubation period (Table 1, Fig. S15).

## 4 Discussion

### 4.1 Environmental conditions driving higher $CO_2$ emissions in the fen

During the incubation, we measured $CO_2$ emissions that were three to four times higher in the fen compared to the palsa when the cores were incubated under in-situ moisture conditions (Fig. 5a, Table 1). Our results align with previous incubation studies
showing two to four times higher $CO_2$ production rates from fens than from palsas (Baysinger et al., 2025; Kirkwood et al., 2021; Kjær et al., 2024). However, since we removed vegetation from the top layer, the $CO_2$ emissions we measured were likely higher than under true in-situ conditions, especially for the fens, which typically act as a C sink during the growing season (Holmes et al., 2022; Hugelius et al., 2020). For comparison, Voigt et al. (2017) found that seasonal CO emissions were about three times higher under bare conditions than under vegetated conditions in a warmed treatment. Under control
conditions, vegetated sites acted as a $CO_2$ sink during July and August, while bare sites remained a $CO_2$ source throughout the snow-free season.

The accumulation and decomposition of peat at the studied sites are shaped by distinct hydrological conditions. The fen site has been permafrost-free for approximately 25 years, in the uppermost 2.5 meters (since 2000), during which the top 23 cm of peat has accumulated (Fig. 1; Reif (2023); Verdonen et al. (2023)). This layer consists of modern C inputs from fresh vegetation.
Under water-saturated conditions, decomposition is limited by the lack of terminal electron acceptors (TEAs) (Reddy and DeLaune, 2008), which leads to a higher accumulation of OM (Lee, 1992; Turetsky et al., 2000). In contrast, the active layer of the palsa contained OM deposited over the last 4,600 years and was exposed to decomposition due to dry conditions (Reif, 2023). This prolonged exposure to decomposition in the active layer of the palsa reduced the potential C loss from this site over time (Treat et al., 2021). Our FTIR results indicate that the OM in the active layer of the palsa, prior to incubation, is more
decomposed than the OM in the first 60 cm of the fen, reflecting the different hydrological conditions (Fig. 4a) and OM inputs. The dry soil conditions in the palsa led to a shrub-dominated environment with a dried *Sphagnum* layer at the surface. This type of vegetation is typically associated with acidic soils, (pH <4), and our pH results (from 3.4 to 3.6) for the active layer fall into the range of magnitude of palsas (Hodgkins et al., 2014; Kirkwood et al., 2021; Treat et al., 2015). However, *Sphagnum* moss is difficult to decompose because it captures carbohydrates and releases organic acids that inhibit decomposition, which
decreases the lability of OM (Mellegård et al., 2009; Zaitseva et al., 2010). The shift from *Sphagnum* to sedge-dominated peat in the fen caused an increase in pH that facilitates OM degradation. Additionally, we measured DOC concentrations that were 2.5 times higher (up to 500 mg.$L^{-1}$) in the first 20 cm of the fen core (Fig. 3, Fig. S21). DOC from sedge-dominated peat is more labile than DOC from shrub-dominated peat (Chanton et al., 2008). Therefore, we observed a surface layer in the fen with higher and more labile DOC, which could explain the higher $CO_2$ emissions measured during the incubation.
To summarize, although aerobic conditions facilitate OM decomposition in the palsa, the high degree of peat decomposition combined with low OM inputs has resulted in more recalcitrant OM in the active layer of the palsa (Fig. 4a), potentially limiting further C mineralization (Harris et al., 2023). In contrast, while the fen is $O_2$-limited, the vegetation, fresh OM input, and adapted microbial communities led to threefold higher $CO_2$ emissions.

## 4.2 Increase in $CO_2$ emissions after flooding, simulation of abrupt thaw

In our experimental setting, abrupt thaw simulations resulted in a 55% increase in $CO_2$ emissions under flooded conditions compared to gradual thaw simulations (Control) (Table 1, Fig. 6). These results indicate that the abrupt flooding scenario had a greater impact on $CO_2$ emissions than permafrost thaw alone. This contrasts with previous incubation studies simulating permafrost thaw in palsa, where lower $CO_2$ production was observed under anaerobic conditions (Baysinger et al., 2025; Treat et al., 2014). However, Voigt et al. (2019), in their mesocosm study, did not observe a significant difference in $CO_2$ emissions between flooded and in-situ moisture conditions. Under in-situ moisture conditions, the first 60 cm of the palsa remained aerobic. Therefore, even under flooded conditions, the meso-scale of our experiment delayed the establishment of anaerobic conditions compared to traditional incubations. After a three-month incubation, the flooded palsa mesocosms at 10°C had not reached anaerobic conditions, which also explains the absence of $CH_4$ emissions. Under field conditions, chronosequence studies have shown that $CH_4$ emissions increase during the young and intermediate thaw stages (approximately 10-60 years after thaw), when soils are wetter and vegetation is adapted to these conditions. The lack of anaerobic conditions in our mesocosms likely reflects the short experimental duration compared to natural thaw dynamics (three months vs. several decades). As bogs mature and peat accumulates, the water table lowers, leading to drier conditions and a subsequent decline in $CH_4$ emissions (Johnston et al., 2014; Heffernan et al., 2022, 2024). Additionally, higher $CO_2$ emissions under flooded conditions have also been observed in field studies during the first years after thaw (< 4 years) (Kuhn et al., 2018; Rodenhizer et al., 2023a), supporting the potential for increased $CO_2$ emissions with higher moisture content when $O_2$ remains available.

In their review paper, Kim et al. (2010) compiled studies from various experimental setups and soil types to explain the observed increase in $CO_2$ emissions under flooded conditions. They attributed this increase to higher C lability during flooding events, driven by several mechanisms: (1) changes in soil redox properties (Chen et al., 2020; Patzner et al., 2020), (2) vertical substrate mobilization (Xiang et al., 2008), (3) increased microbial biomass C (Clein and Schimel, 1994; Fierer and Schimel, 2003), and (4) physical disruption, which makes soil OM more accessible (Denef et al., 2001). However, the latter mechanism is likely a minor contributor in peat soils, due to their low mineral content, which limits aggregate formation (Han et al., 2016). The sudden increase in soil moisture with flooding changes the availability of TEAs, changing the redox status and reducing the likelihood of $CH_4$ production. Some palsas are iron-rich environments, which can help protect OM from decomposition under aerobic conditions but under anaerobic conditions, Fe(III) can be reduced to aqueous Fe(II), thereby making OM more accessible for microbial decomposition (Chen et al., 2020; Patzner et al., 2020). However, other more favorable TEAs such as $NO_3^-$ could be used first to decompose OM and therefore delay the utilization of Fe(III).

Furthermore, flooding stimulates vertical transport within the soil column, mobilizing OM and nutrients from deeper layers to the surface, which can enhance C mineralization (Xiang et al., 2008). In both scenarios, we observed the greatest increase in OM decomposition at the permafrost table (Fig. 4b). However, decomposition was more pronounced under flooding conditions (DI: 1.9 vs. 1.3 in flooded and control treatments, respectively; Fig. 4b). With enhanced vertical transport, freshly decomposed substrates can be mobilized to upper soil layers (Tfaily et al., 2018).

Rewetting soil can also lead to microbial cell lysis, releasing nutrients and increasing microbial biomass (Clein and Schimel,

1994; Fierer and Schimel, 2003; Kim et al., 2012). Most studies reporting $CO_2$ increases due to cell lysis have observed a short $CO_2$ pulse lasting up to three days following the rewetting event (Clein and Schimel, 1994; Fierer and Schimel, 2003; Kim et al., 2012). Additionally, because we collected our cores during winter, both the soil and microbial communities were subject to freeze–thaw disruption, which could induce thermal shock and amplify cell lysis. In our study, $CO_2$ released from microbial biomass could partly explain the pulses observed during the first two to three days of incubation, rather than the overall elevated emissions (Fig. 5). $CO_2$ pulses following thaw could also be related to releases of trapped $CO_2$. Finally, these $CO_2$ pulses may also be enhanced by physical disturbances due to sequential thawing and, therefore, may not represent natural processes. However, since all the mesocosms were thawed under similar conditions, potential releases of trapped $CO_2$ or physical disturbances due to setup limitations should be comparable across mesocosms (excluding micro-scale spatial heterogeneity).

The lack of significant differences between water treatments during incubation (Fig. 6, Table 1) is likely due to the overriding effect of water saturation itself, which may have superseded any potential influence of varying water compositions. Additionally, the DOC content of the added water was an order of magnitude lower than that of the palsa and fen soils (9.68; Table S3), thereby reducing the impact of waterborne OM on $CO_2$ emissions. Baysinger et al. (2025) reported a 60% decrease in $CO_2$ production in the permafrost layer under the non-filtered treatment compared to sterile water over a 384-day incubation, likely due to the establishment of anaerobic conditions and the absence of vertical transport. In contrast, our results consistently show approximately twofold higher $CO_2$ emissions under flooded conditions. We attribute this primarily to vertical substrate mobilization, residual $O_2$ within the soil column, or redox shifts driven by increased water content, rather than to microbial or chemical inputs from thermokarst pond water. Nonetheless, methodological factors—such as the lateral mode of water addition and data filtering—may have influenced the observed treatment effects.

Finally, we observed $CH_4$ uptake ranging from 15 to 34.6 $\mu$g-$CH_4$ m$^{-2}$ h$^{-1}$ (Table 1, Fig. S15), even under flooded conditions — consistent with findings by Voigt et al. (2019), who reported uptake ranging from 0.35 to 2.42 mg-$CH_4$ m$^{-2}$ d$^{-1}$. This observation aligns with results from smaller incubation studies that reported little to no $CH_4$ production in the active layer (Baysinger et al., 2025; Kirkwood et al., 2021; Treat et al., 2014). We attribute the observed $CH_4$ uptake to methane oxidation within the soil column, likely facilitated by the presence of residual $O_2$. Under in situ conditions, the active layer tends to remain dry and oxygenated, which inhibits $CH_4$ production.

Palsas in Fennoscandia experience degradation rates that vary between -1.0 % yr$^{-1}$ and -1.3 % yr$^{-1}$ over the period of 1950/60-2010/2014 (Borge et al., 2017; Leppiniemi et al., 2025; Olvmo et al., 2020). Palsa collapse has been identified as a major driver of permafrost degradation (Verdonen et al., 2023), highlighting the importance of abrupt thaw. While our study focuses on short-term $CO_2$ responses (three-month incubation) driven by hydrological differences between thaw types, these processes also will likely differ in their temporal dynamics. Our results suggest that $CO_2$ emissions from abrupt thaw peak early and then stabilize and/or decrease as anaerobic conditions develop, whereas emissions from gradual thaw may increase more steadily as deeper layers thaw. The rise in $CO_2$ emissions from the Control during the final 10 days of incubation may reflect this slower process (Table 2). Over longer timescales, palsas undergoing thaw (both abrupt and gradual) transition from initial $CO_2$ sources to net sinks as vegetation re-establishes and ecosystem productivity increases, a process that may take several decades (Johnston et al., 2014; Heffernan et al., 2022, 2024). This process is difficult to capture accurately under field conditions,

highlighting the importance of continuous, long-term monitoring to assess $CO_2$ emissions during organic matter degradation – particularly before vegetation shifts and the onset of $CH_4$ emissions.

### 4.3 Comparison of Deeper Soil Contributions: Permafrost Degradation Simulation vs. Stabilized Thawed fen

C mineralization following permafrost thaw from peat partly depends on the stage of peat decomposition (Estop-Aragonés et al., 2022). Our FTIR results, along with soil parameters measured prior to incubation, indicate a higher potential for OM decomposition in the permafrost layer (>60 cm) compared to the active layer (0-60 cm), as evidenced by higher pH values
and reductions in CH and C=O peaks (Fig. 3, Fig. 4a). According to the post-incubation FTIR spectra and DI ratios, the highest decomposition occurred at the permafrost table (60 cm) under both flooded and dry conditions (Fig. 4b).This enhanced decomposition at the permafrost table can be attributed to the presence of active microbial communities—likely translocated from the active layer during thaw—and the availability of relatively undecomposed peat. In contrast, little to no peat decomposition occurred deeper within the permafrost layer (60–100 cm; Fig. 4b), despite the observed increase in $CO_2$ emissions
under control conditions (Table 2). Furthermore, we observed minimal differences in the stage of peat decomposition between the deeper permafrost layer and the equivalent layer in the fen (60–100 cm; Fig. 3, Fig. 4a). This suggests that both the short-term thaw simulated in this study and 25 years of permafrost-free conditions have not significantly altered OM quality in the deeper permafrost layers (>60 cm). These findings are consistent with observations from other permafrost regions, such as in Canada (Harris et al., 2023). Thus, we conclude that peat degradation during permafrost thaw is primarily concentrated at the
permafrost table, whereas in deeper layers, neither thaw processes nor long-term permafrost loss have notably affected peat quality.

Reif (2023) estimated that the fen site in Peera lost approximately 15 kg C m$^{-2}$ since thaw, without indicating any significant increase in peat decomposition at the fen site (Fig. 4a; Reif (2023)). This finding is consistent with Harris et al. (2023), who measured a net C loss of about 19.3 kg C m$^{-2}$ since thaw, with no observable changes in peat quality. Several studies have re-
ported lower $CO_2$ emissions from degrading permafrost under waterlogged conditions (Estop-Aragonés et al., 2018a, b; Voigt et al., 2019), which could explain the minimal change in peat quality observed in the fen. This is in line with our observation of decreased $CO_2$ emissions per volume of thawed soil in the abrupt thaw simulation (flooded) and higher emissions in the gradual thaw simulation (Control) during the final thaw step (0–100 cm; Table 2). The greater contribution from deeper layers under control conditions may result from increased gas diffusivity in the drier soil column. However, without $^{14}C$ analysis, it
remains unclear whether emissions originate solely from the 80–100 cm layer or if the 60–80 cm layer contributed later due to lower diffusivity. Under flooded conditions, water saturation limits oxygen availability and slows gas diffusion (Voigt et al., 2019), leading to anaerobic conditions within the permafrost layer. Although microbial decomposition still occurs, it shifts to less efficient anaerobic pathways that produce $CO_2$ (along with other gases), which then accumulates due to restricted diffusion through the saturated soil.
In the fen mesocosms, $CO_2$ emissions increased with depth despite flooded conditions (Table 1, Fig. 6), with the highest DIs observed at the bottom of the cores (Fig. 4b). While surface vegetation was removed during incubation, the root systems were

retained. The presence of deep-rooted aerenchymous plants, such as sedges, may explain the increased $CO_2$ emissions in the fen. Aerenchyma tissue enhances the transport of $O_2$, $CO_2$, and $CH_4$ in waterlogged soils (Armstrong, 1980; Colmer, 2003). Although aerenchyma are best known for enhancing $CH_4$ transport and reducing $CH_4$ oxidation, they can also provide pathways for $CO_2$ movement. Because $CO_2$ diffuses roughly 10,000 times faster in air than in water (Armstrong, 1980; Colmer, 2003; Tiner, 2005), gas exchange through aerenchyma can substantially influence $CO_2$ fluxes. In this process, $O_2$ is transported downward to the roots, while $CO_2$ produced by microbial activity moves upward and is fixed in the shoots via photosynthesis (Dacey and Klug, 1982; Pedersen and Sand-Jensen, 1992; Smith and Russell, 1969; Smith et al., 1983). In the absence of photosynthesis, roots may have acted as pathways for $CO_2$ from deeper layers. However, the artificial sequential thawing process may have amplified these emissions, leading to potential overestimation of $CO_2$ fluxes.

Finally, the degradation of dead roots could provide additional substrate, contributing to the higher DI observed at the bottom layer (Fig. 4b). This process may also increase soil porosity, facilitating gas exchange to the surface. This highlights the potential role of vegetation change since thaw in $CO_2$ transport, particularly when there is no active vegetation. While several studies have measured and modeled increased $CO_2$ emissions in northern fens during the non-growing and shoulder seasons (Fahnestock et al., 1999; Joiner et al., 1999; Natali et al., 2019; Rafat et al., 2021), few have accounted for the influence of vegetation type and root structure on $CO_2$ dynamics. This represents a significant gap in existing research and models, underscoring the need for further investigation in this area.

## 5   Conclusion

In conclusion, our findings contradicted our initial hypothesis by showing that the abrupt thaw simulation (i.e., flooding treatments) significantly enhanced $CO_2$ emissions with permafrost thaw, even when microbial communities had not yet adapted to saturated conditions (H1). This suggests that vertical C transport, residual $O_2$, and redox changes induced by flooding play a crucial role in $CO_2$ dynamics during the thaw transition in abrupt palsa permafrost. By contrast, the composition of thermokarst thaw water appears to contribute less to C mineralization than the sudden increase in soil moisture (H2). Consistent with other studies, we observed an increase in $CO_2$ emissions only under gradual thaw simulations in thawing permafrost horizons. Additionally, $CO_2$ emissions from fens increased under water-saturated conditions (H3), possibly reflecting $CO_2$ transport from deeper soil layers mediated by deep-rooted vegetation during non-growing and shoulder seasons. Our pilot study underscores that substantial $CO_2$ emissions can be released during permafrost thaw, even under water-saturated conditions. Future research should focus on quantifying $CO_2$ release during abrupt thaw under in situ conditions. Additionally, model-based future projections should account for the potential contribution of greenhouse gases from deeper layers and the role of deep-rooted vegetation in producing and transporting $CO_2$ outside the growing season.

*Data availability.*   The data sets used in this paper are available at doi: 10.1594/PANGAEA.974304 and doi: 10.1594/PANGAEA.974302

*Author contributions.* M.L., C.T. and J.Sch. designed the study. M.L., J.St. and T.W. collected the samples during the expedition in 2023. M.L. built the incubator and conducted the experiment, supported by M.Lü. M.H. and R.E. provided the laboratory infrastructure and the original script for the flux calculations. M.L. carried out the data analyses and M.H. provided the script. M.B. assisted with data interpretation.
535 M.L. wrote the manuscript with contributions from all the co-authors.

*Competing interests.* The authors declare no conflict of interest.

*Acknowledgements.* This study was funded by the Alfred Wegener Institute (grant details: INSPIRES PhD Fellowship) and the European Research Council (ERC grant: FluxWIN Project 851181). We are grateful to the Research Station of Kilpisjärvi for their assistance during fieldwork. We would like to acknowledge J. Vollmer, M. Dolle and S. Wocheslander, as well as the AWI lab technicians, for their support with
540 laboratory work. We also thank our colleagues for their valuable discussions and feedback. Finally, we extend our gratitude to the reviewers whose comments helped improve this manuscript.

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
