# Peer review of "Enhanced CO2 Emissions Driven by Flooding in a Simulation of Palsa Degradation"

_EGUsphere, 2025_

## Referee Comment (RC1)

Review of "Enhanced CO2 Emissions Driven by Flooding in a Simulation of Palsa Degradation"

**General comments:**

The study presented here provides meaningful insights in the role of water and peat quality upon permafrost thaw, which is I highly relevant and recent research topic. BG seems to be an appropriate journal for the publication of this study.

Mesocosm incubations are a robust and established method, although the simulation of permafrost thaw via partial freeing and thawing is technically demanding and therefore only few studies exist, which the authors also correctly emphasize. The study is therefore addressing a recent and relevant research gap. The authors describe their experiments clear and concise, although open questions regarding the choice of water level, experimental time frame and data filtering remain, which are more specifically addressed in the specific comments. Briefly, the assumption of a certain water level that stays constant over the chosen time period is realistic but just one of many scenarios. Therefore, the manuscript would profit from a better justification of why these parameters were chosen the way they are. An important issue is the filtering of data: The authors state that due to saturation of the sensor, no CO2 values greater than 5.000 ppm could be accurately measured and were filtered out. However, the presentation of the results does not allow to understand the extent of this filtering and how it affects the overall results and statistics. Furthermore, the manuscript lacks any hypotheses and only a general aim is stated, which should be clarified. The conclusion section mentions an initial hypothesis which is contradicted by the results but there is no such hypothesis stated in the introduction. Generally, the objectives are rather short and it seems like this study was performed more explorative rather than having clear expectations or hypotheses. The actual difference between "abrupt" and "gradual" thaw needs also more explanation, since it is not very clear how fast abrupt thawing is defined in this study.

Overall, the paper needs some clarification and justifications but seems to be suitable for publication after addressing the concerns mentioned above and in the specific comments. The title and abstract are appropriate and the overall language and presentation are well chosen. The authors take the recent literature into account and summarize it sufficiently to understand the research gaps and limitations of the methodology.

**Specific comments:**

L 29f: Please specify the kinds of changes (e.g. how will the vegetation change, will it become wetter or drier, etc.)

L 42ff: What is meant by "C production"? Shouldn't it be gas production? Also, consider changing "C decomposition" to "OM decomposition"

L 100: What was sampled in October? Or was it just an exploratory visit in order to map vegetation and active layer depth?

L 110: Please specify how the corer was modified.

L 131ff: How fast was that abrupt thaw?

L 173ff: Fluxes will be underestimated when all flux data > 5000 ppm $CO_2$ is filtered out. With this in mind, results can still be interpreted in some way but it would be helpful to have information about the timing of these extreme values. Where they equally distributed throughout the experiment or

did this problem occur only during a specific time frame? This information could be included in a graphic like figure 5 or figure S14 – S18

L 240ff: That was already explained in section 2.5.1.

L 249ff: Did you check beforehand whether the criteria for the tests (normality, homogeneity in variance, etc.) were fulfilled?

L 347f: Can you provide a rough estimate of how much higher your emissions are compared to other studies?

L 361: It would be good to have the measured and typical pH values stated here.

L 364: Did you also measure DOC after your incubation experiment? It would be interesting to see some kind of mass balance of OC over the incubation to get an idea of decomposition pathways, e.g. to see how much solid OC and DOC are transferred to gases and vice-versa.

L 381: Reads like there is a big bias caused by the dimension of the samples? Would a real-world scenario then maybe never reach anaerobic conditions because the soil dries or refreezes before?

L 391ff: I agree with the general concept of carbon release upon Fe(III) reduction and that this mechanisms can (partly) explain the results found here. However, I question that palsas are always Fe-rich. The cited work was a case-study and it would be good to see some kind of comparison of the both sites in terms of palsa formation, underlaying geology, etc. The addition of water indeed hampers oxygen availability but before Fe is reduced, other TEAs ($NO_3$, $MnO_2$) are used, which also needs some time. Since this study did not find anaerobic conditions immediately, couldn't it be that Fe is still not reduced?

L 484: How realistic is this abrupt thaw scenario? Since it was not stated how long it takes under that scenario, it is hard to estimate whether this is just a theoretical scenario or realistic in permafrost regions.

**Technical corrections:**

L 1: Remove comma

L 44: Is Baysinger still in prep? This study is cited quite often here, which is a bit unfortune when it is still in preparation .

L 247: Parenthesis before the phrase is not necessary

L. 256: R Core team not in literature list

L 260: Check the wording. "Deepest value" is slightly confusing since it also seems to be the lowest value? Do you mean the sample at the bottom of the core?

L 478: Missing space between "C" and "transport"

---

## Author Comment (AC1)

**General response:**

We thank the reviewers for their time and insightful remarks which will help improving the manuscript. Both reviewers commented on the simulation of abrupt versus gradual thaw. We will clarify the definition of abrupt vs. gradual thaw based on the review paper from Webb et al. 2025 and add additional information about rates of palsa thaw based on field studies from northern Fennoscandia to constrain the time scales of flooding. We will also clarify some methodological points (see specific comments) and add statistical comparison before and after the quality control to unsure the robustness of our results. We will also include in the discussion the specific comments from both reviewers such as  $CO_2$  trapped in the permafrost layer or the temporal dimension between abrupt and gradual thaw.

Please, not that the line numbers refer to the initial manuscript.

**Reviewer 1:**

The study presented here provides meaningful insights in the role of water and peat quality upon permafrost thaw, which is I highly relevant and recent research topic. BG seems to be an appropriate journal for the publication of this study.

Mesocosm incubations are a robust and established method, although the simulation of permafrost thaw via partial freeing and thawing is technically demanding and therefore only few studies exist, which the authors also correctly emphasize. The study is therefore addressing a recent and relevant research gap.

The authors describe their experiments clear and concise, although open questions regarding the choice of water level, experimental time frame and data filtering remain, which are more specifically addressed in the specific comments. Briefly, the assumption of a certain water level that stays constant over the chosen time period is realistic but just one of many scenarios. Therefore, the manuscript would profit from a better justification of why these parameters were chosen the way they are.

Authors' Response (AR): We chose to flood the mesocosms under abrupt conditions up to the surface to match the water table depth at the fen site (see picture of the fen; figure 2). We now clarified this in 1.135:

"To simulate the sudden increase in soil moisture during abrupt thaw, we flooded the mesocosms until the water table reached the surface (0 cm). We chose to flood the mesocosm with a similar water table as the one measured at the fen site to be able to compare emissions from both sites."

Regarding the experimental time frame, in l. 123 we explain the mesocosms were incubated for 12 weeks, and give more details regarding the sequential thawing in section 2.3.4. Based on your comment l. 131ff, we will clarify the definition of abrupt and gradual thaw in section 2.3.2 (see AR to the mentioned comment).

An important issue is the filtering of data: The authors state that due to saturation of the sensor, no CO2 values greater than 5.000 ppm could be accurately measured and were filtered out. However, the presentation of the results does not allow to understand the extent of this filtering and how it affects the overall results and statistics.

AR: Based on your specific comment L. 173ff, we will add in Fig. S18 the distribution of data deleted over time. Overall, 11.7 % of the data were filtered, we will edit Table S1 and add the total percentage of data filtered during the quality control. Additionally in Fig. 19 we show the distribution of the full data set without the filtering. However, we agree that in the manuscript it is not clear how the filtering affected the statistics. Therefore, we suggest to do the statistics on the non-filtered dataset and to include the results in the Supplementary. Because extreme values strongly affect mean-based comparisons, we employed robust, median-based statistical methods that are less sensitive to outliers to compare the two

datasets (with and without filtering). These include a Kruskal-Wallis test followed by pairwise comparisons with Dunn's test. We propose to present the results of these analyses in the supplementary to show that filtering did not influence the statistics and overall results. We will include this analysis in the method and explain why we used median instead of mean to test the robustness of the results. If needed, we can also add a scatter plot showing the median over each thaw step before and after quality control.

Table 1: Comparison of Kruskal–Wallis post-hoc test results before and after QC (adjusted p-values). "sig." indicates significant after multiple testing correction ( $\alpha$  = 0.05). Note that the numbers in the comparison list indicate the depth in cm.

| Comparison                                 | After QC | Before QC | Outcome    |
|--------------------------------------------|----------|-----------|------------|
| Control_100_Palsa – Control_60_Palsa       | n.s.     | n.s.      | consistent |
| Control_100_Palsa – Control_80_Palsa       | n.s.     | n.s.      | consistent |
| Control_60_Palsa - Control_80_Palsa        | n.s.     | n.s.      | consistent |
| Control_100_Palsa - Flooded_100_Palsa      | n.s.     | n.s.      | consistent |
| Control_60_Palsa – Flooded_100_Palsa       | n.s.     | n.s.      | consistent |
| Control_80_Palsa – Flooded_100_Palsa       | n.s.     | n.s.      | consistent |
| Control_100_Palsa – Flooded_100_Peatland   | n.s.     | n.s.      | consistent |
| Control_60_Palsa – Flooded_100_Peatland    | sig.     | sig.      | consistent |
| Control_80_Palsa – Flooded_100_Peatland    | sig.     | sig.      | consistent |
| Flooded_100_Palsa - Flooded_100_Peatland   | n.s.     | n.s.      | consistent |
| Control_100_Palsa – Flooded_60_Palsa       | n.s.     | n.s.      | consistent |
| Control_60_Palsa – Flooded_60_Palsa        | sig.     | sig.      | consistent |
| Control_80_Palsa – Flooded_60_Palsa        | sig.     | sig.      | consistent |
| Flooded_100_Palsa - Flooded_60_Palsa       | n.s.     | n.s.      | consistent |
| Flooded_100_Peatland - Flooded_60_Palsa    | n.s.     | n.s.      | consistent |
| Control_100_Palsa – Flooded_60_Peatland    | n.s.     | n.s.      | consistent |
| Control_60_Palsa – Flooded_60_Peatland     | n.s.     | n.s.      | consistent |
| Control_80_Palsa – Flooded_60_Peatland     | n.s.     | n.s.      | consistent |
| Flooded_100_Palsa - Flooded_60_Peatland    | n.s.     | n.s.      | consistent |
| Flooded_100_Peatland - Flooded_60_Peatland | n.s.     | n.s.      | consistent |
| Flooded_60_Palsa – Flooded_60_Peatland     | n.s.     | n.s.      | consistent |
| Control_100_Palsa – Flooded_80_Palsa       | n.s.     | n.s.      | consistent |
| Control_60_Palsa – Flooded_80_Palsa        | n.s.     | n.s.      | consistent |
| Control_80_Palsa – Flooded_80_Palsa        | n.s.     | n.s.      | consistent |
| Flooded_100_Palsa - Flooded_80_Palsa       | n.s.     | n.s.      | consistent |
| Flooded_100_Peatland - Flooded_80_Palsa    | sig.     | sig.      | consistent |
| Flooded_60_Palsa - Flooded_80_Palsa        | sig.     | sig.      | consistent |
| Flooded_60_Peatland - Flooded_80_Palsa     | n.s.     | n.s.      | consistent |
| Control_100_Palsa – Flooded_80_Peatland    | n.s.     | n.s.      | consistent |
| Control_60_Palsa – Flooded_80_Peatland     | sig.     | sig.      | consistent |

| Comparison                                 | After QC | Before QC | Outcome    |
|--------------------------------------------|----------|-----------|------------|
| Control_80_Palsa – Flooded_80_Peatland     | sig.     | sig.      | consistent |
| Flooded_100_Palsa - Flooded_80_Peatland    | n.s.     | n.s.      | consistent |
| Flooded_100_Peatland - Flooded_80_Peatland | n.s.     | n.s.      | consistent |
| Flooded_60_Palsa - Flooded_80_Peatland     | n.s.     | n.s.      | consistent |
| Flooded_60_Peatland - Flooded_80_Peatland  | n.s.     | n.s.      | consistent |
| Flooded_80_Palsa - Flooded_80_Peatland     | sig.     | sig.      | consistent |

Furthermore, the manuscript lacks any hypotheses and only a general aim is stated, which should be clarified. The conclusion section mentions an initial hypothesis which is contradicted by the results but there is no such hypothesis stated in the introduction. Generally, the objectives are rather short and it seems like this study was performed more explorative rather than having clear expectations or hypotheses

AR: Thank you for pointing this out. We agree that our initial hypotheses should be added in the introduction line 88:

"Based on this design, we hypothesized that (H1) flooding with abrupt thaw alters vertical C dynamics by physically disturbing the soil matrix and changing the redox properties of the soil to anaerobic conditions. Anaerobic conditions under the abrupt thaw simulations result to lower CO2 emissions compared to gradual thaw. (H2) The addition of thermokarst pond water during the thaw simulation enhances C emissions through increase microbial activity due to microbial colonization from the thermokarst pond water. (H3) Long-term thaw in the fen results in higher C emissions than in the palsa because of fresh organic matter input, permafrost-free conditions for several decades and therefore already established microbial communities. The in-situ water logged conditions in the fen site allow for CH4 emissions during the incubation, while under abrupt thaw simulation, ideal redox conditions are not reached after three-month incubation despite water saturation and therefore no CH4 emission occurs."

We will rephrase the conclusion accordingly by mentioning comparing our findings to the initial hypotheses.

The actual difference between "abrupt" and "gradual" thaw needs also more explanation, since it is not very clear how fast abrupt thawing is defined in this study.

AR: In the introduction, we define abrupt thaw as thaw that can happen "abruptly within days to years, with the latter often forming thermokarst ponds at collapsing palsa edges" (l. 67) and highlight the lack of consideration of hydrology during abrupt thaw. A recent review paper (Webb et al., 2025) published while this paper was in review redefines abrupt thaw as a thaw feature that emerges within less than 30 years with an ice-content >20% and/or with a large impact on the ecosystem/complete stage change (e.g., wildfire, streamflow). With this definition, abrupt thaw is not only defined across a temporal component, but also across environmental and soil properties such as hydrology and sudden increase in soil moisture. We will clarify the definition in the introduction to better justify why in our abrupt vs. gradual thaw simulation we applied the same thawing process. To clarify the definition of abrupt thaw and better constrain the timescale of thawing, we will indicate palsa degradation rates in Fennoscandia.

**Suggestion:**

"Thaw can proceed gradually over decades or abruptly within days to years, with the latter often forming thermokarst ponds at collapsing palsa edges (Quinton and Baltzer, 2013; Jorgenson et al., 2006; Borge

et al., 2017). Abrupt thaw has recently been more precisely defined as occurring within 30 years in ice-rich soils (>20%) and/or when causing major ecological or state shifts such as wildfire or streamflow changes (Webb et al., 2025). This definition incorporates not only a temporal criterion but also environmental and soil factors such as hydrology and rapid soil moisture increase. With this updated definition, abrupt thaw appears to be the dominant form of permafrost degradation in palsas. The ice-rich permafrost layer in palsas leads to ground subsidence and hydrological changes, driving the transition from elevated palsas to wetter peatland ecosystems such as bogs and fens (Hugelius et al., 2020). Additionally, studies from Fennoscandia estimated palsa degradation rates between -1.0% yr-1 and -1.3% yr-1 over the period of 1950/60-2010/2014 (Borge et al. 2017, Leppienemi et al. 2025, Olvmo et al., 2020) with palsa losses reaching up to 80 % between 2007 and 2021 in specific areas such as Finnish Lapland. Permafrost degradation was primarily attributed to palsa collapse rather than active layer deepening (Verdonen et al. 2023)."

Overall, the paper needs some clarification and justifications but seems to be suitable for publication after addressing the concerns mentioned above and in the specific comments. The title and abstract are appropriate and the overall language and presentation are well chosen. The authors take the recent literature into account and summarize it sufficiently to understand the research gaps and limitations of the methodology.

**Specific comments:**

L 29f: Please specify the kinds of changes (e.g. how will the vegetation change, will it become wetter or drier, etc.)

AR: With the loss of palsas, the soil moisture will increase with the formation of wetlands. Following this, the shrub-like vegetation will switch to deep-rooted plants such as sedges (Hugelius et al., 2020; Malhotra & Roulet, 2015). We will include this in the manuscript: "The loss of palsas will switch vegetation to deep-rooted plants such as sedges, change hydrology by increasing soil moisture."

L 42ff: What is meant by "C production"? Shouldn't it be gas production? Also, consider changing "C decomposition" to "OM decomposition"

AR: We will rephrase "C production" by "gas (CO2 and CH4) production". We change "C decomposition" to "OM decomposition".

L 100: What was sampled in October? Or was it just an exploratory visit in order to map vegetation and active layer depth?

AR: In October 2022 we collected soil samples for three other studies. However, for this manuscript, we only used the site description data such as active layer depths, vegetation, water table depth, etc. We agree that the phrasing is a bit confusing and suggest to change it to "The active layer was measured during the first exploratory expedition in October 2022".

L 110: Please specify how the corer was modified.

AR: Thank you. Actually, the SIPRE auger was not modified. We will delete this term.

L 131ff: How fast was that abrupt thaw?

AR: For all the treatments, we thaw the permafrost layer (40cm) over three months of incubation. In 1. 149ff we explain that all the mesocosms were thawed at the same speed to isolate the effect of hydrological changes between abrupt vs. thaw. To make this clearer, we suggest to add a sentence in

the section 2.3.2: "For both thaw simulations (gradual and abrupt thaw) mesocosms were thawed at the same speed and only the water content was modified."

L 173ff: Fluxes will be underestimated when all flux data > 5000 ppm  $CO_2$  is filtered out. With this in mind, results can still be interpreted in some way but it would be helpful to have information about the timing of these extreme values. Where they equally distributed throughout the experiment or did this problem occur only during a specific time frame? This information could be included in a graphic like figure 5 or figure S14-S18

AR: Thank you for this remark. To show more clearly the distribution of the extreme values we will add a panel under each existing panel in Fig. S18. The new panels will display the distribution of the anomalies over time for each replicate using a ridge plot visualization (see example fig below).

Figure 1: Example for the control treatment of the revised S18 figures. Panel i) shows the CO2 flux timeseries of each replicate the vertical orange lines indicate the thaw stages and the dashed blue lines indicate the water addition times. and ii) shows the density of measured CO2 saturation events over\_time. The numbers on the left refer to the channel's numbers and are identical to the ones from Table S1. Colors differentiate replicates.

L 240ff: That was already explained in section 2.5.1.

AR: Thank you very much for notifying this. We will delete the section 2.5.4.

L 249ff: Did you check beforehand whether the criteria for the tests (normality, homogeneity in variance, etc.) were fulfilled?

AR: The data were tested for normality, homogeneity of variance and independency within groups. We will add this information in section 2.6. by stating "Before performing statistical test, we tested the data for independency, normality and homogeneity of variance."

L 347f: Can you provide a rough estimate of how much higher your emissions are compared to other studies?

AR: As we did not have a treatment with vegetation, we cannot provide an estimate for this specific site. However, in previous in-situ studies, Voigt et al. (2017) measured a seasonal CO2 balance for the whole snow-free season 3 times higher under bare conditions than under vegetated conditions (warmed treatment). Under control conditions, the vegetated site was a net CO2 sink during July and August while the bare site was acting as a CO2 source during the whole snow-free season. We will add this comparison to the discussion (after L 347f) to give an order of magnitude, while specifying that CO2 budget is strongly related to the site conditions.

**Suggestion:**

"For comparison, Voigt et al. (2017) found that seasonal CO2 emissions were about three times higher under bare conditions than under vegetated conditions in a warmed treatment. Under control conditions, vegetated sites acted as a CO2 sink during July and August, while bare sites remained a CO2 source throughout the snow-free season."

L 361: It would be good to have the measured and typical pH values stated here.

AR: We will edit the text to read: "This type of vegetation is typically associated with acidic soils (pH <4), and our pH results (from 3.4 to 3.6) for the active layer fall into the range of magnitude of palsas (from 3.2 to 4.8; Hodgkins et al., 2014; Kirkwood et al., 2021; Treat et al., 2015)."

L 364: Did you also measure DOC after your incubation experiment? It would be interesting to see some kind of mass balance of OC over the incubation to get an idea of decomposition pathways, e.g. to see how much solid OC and DOC are transferred to gases and vice-versa.

AR: We measured DOC prior to each thaw step at four depths (0–2 cm, 25 cm, 50 cm, and 75 cm; when the soil was thawed). However, because water was not always available or in too little quantity, we initially decided not to include these data. We now propose adding a plot of the measured DOC values over time in the Supplementary Information and discussing the temporal changes in DOC. Nevertheless, since DOC values were not obtained for all replicates in most treatments, we believe the quality of the data is insufficient to support a robust mass balance of OC.

Figure 2: DOC values after each thawing steps. Most of the depth x incubation time display only one DOC value (representing one replicate) due to too little water sampled.

L 381: Reads like there is a big bias caused by the dimension of the samples? Would a real-world scenario then maybe never reach anaerobic conditions because the soil dries or refreezes before?

AR: Field studies at Peera show where the palsa has completed thawed, conditions are saturated (e.g. Verdonen et al., 2023), so this is likely a transient effect. Furthermore, chronosequence studies along thaw transects (from intact palsa to old bog/fen) show that young to intermediate bogs (10–60 years after thaw) have the highest CH4 emissions due to a high water table and adapted vegetation (Johnston et al. 2014, Heffernan et al. 2021 and Heffernan et al. 2024). As peat accumulates, the water table lowers, and CH4 emissions decline. Field and modeling studies therefore indicate that emissions peak during the early, wetter thaw stages and decrease as the system matures. Since permafrost peatlands are ice-rich and occur in flat terrain, drainage following permafrost thaw is unlikely; soil drying occurs at later stages (>100 years after thaw) due to peat accumulation (Johnston et al. 2014, Heffernan 2021 and Heffernan 2024).

"After a three-month incubation, the flooded palsa mesocosms at 10 °C had not reached anaerobic conditions, which likely explains the absence of CH4 emissions. Under field conditions, chronosequence studies have shown that CH4 emissions increase during the young and intermediate thaw stages (approximately 10–60 years after thaw), when soils are wetter and vegetation is adapted to these conditions. The lack of anaerobic conditions in our mesocosms likely reflects the short experimental duration compared to natural thaw dynamics (three months vs. several decades). As bogs mature and peat accumulates, the water table lowers, leading to drier conditions and a subsequent decline in CH4

emissions (Johnston et al., 2014; Heffernan et al., 2021; Hefferman et al., 2024). Additionally, higher CO2 emissions under flooded conditions have been observed in field studies during the first years after thaw (< 4 years) (Kuhn et al., 2018; Rodenhizer et al., 2023a), supporting the potential for increased CO2 release under higher moisture content when O2 remains available."

L 391ff: I agree with the general concept of carbon release upon Fe(III) reduction and that this mechanisms can (partly) explain the results found here. However, I question that palsas are always Ferich. The cited work was a case-study and it would be good to see some kind of comparison of the both sites in terms of palsa formation, underlaying geology, etc. The addition of water indeed hampers oxygen availability but before Fe is reduced, other TEAs (NO3, MnO2) are used, which also needs some time. Since this study did not find anaerobic conditions immediately, couldn't it be that Fe is still not reduced?

AR: Thank you for this very good remark. We agree that palsas are not always Fe-rich. Based on a metagenomic analysis that was carried as a companion paper of this study, we found that NO3 reduction was a strong metabolic pathway. We will rephrase this sentence to include more favorable reduction pathway.

"The sudden increase in soil moisture with flooding changes the availability of TEAs, changing the redox status and reducing the likelihood of CH4 production. Some palsas are iron-rich environments, which can help protect OM from decomposition under aerobic conditions but under anaerobic conditions, Fe(III) can be reduced to aqueous Fe(II), thereby making OM more accessible for microbial decomposition (Chen et al., 2020; Patzner et al., 2020). However, other more favorable TEAs such as NO3- could be used first to decompose OM and therefore delay the utilization of Fe(III)."

L 484: How realistic is this abrupt thaw scenario? Since it was not stated how long it takes under that scenario, it is hard to estimate whether this is just a theoretical scenario or realistic in permafrost regions.

AR: Based on the updated definition of abrupt thaw by Webb et al. (2025), most palsas can be considered susceptible to abrupt thaw processes. Several studies have quantified palsa loss and subsidence rates across Fennoscandia. In their review, Leppiniemi et al. (2025) reported that Finnish palsas experience faster degradation compared to those in other parts of Fennoscandia, with 63–76% area loss since the 1960s. Reported area loss rates across Fennoscandia range from –1% yr-1 to –3.6% yr-1 (Borge et al., 2017; Leppiniemi et al., 2025; Olvmo et al., 2020; Verdonen et al., 2023), with accelerated loss observed over the past two decades—for instance, the Peera palsa complex lost 55% of its area within fourteen years (Verdonen et al., 2023). We will include palsa degradation rates in Fennoscandia in the introduction (see major comment above) and integrate this point into the discussion.

**Technical corrections:**

L 1: Remove comma

AR: Edited as suggested.

L 44: Is Baysinger still in prep? This study is cited quite often here, which is a bit unfortune when it is still in preparation.

AR: The manuscript is now published and we changed the citations in our manuscript.

L 247: Parenthesis before the phrase is not necessary

AR: Edited as suggested.

L 256: R Core team not in literature list

AR: Thank you for this comment. We will add the citation in the bibliography.

L 260: Check the wording. "Deepest value" is slightly confusing since it also seems to be the lowest value? Do you mean the sample at the bottom of the core?

AR: Thank you for this remark. We agree that the wording is a bit confusing. We changed it with "Depth 100 cm" to clarify that we refer to the sample at the bottom of the core.

L 478: Missing space between "C" and "transport"

AR: Edited as suggested.

**Reviewer 2:**

The presented study is both relevant and of interest to the scientific community, and it fits well within the scope of the journal BG. The mesocosm method is widely used in many studies and is well justified in the framework of this research; detailed experimental schemes and photographs will allow other researchers to reproduce the experiment. It is worth emphasizing the complexity of conducting such experiments and, more generally, the insufficient experimental investigation of the effects of permafrost thaw and soil moisture on CO2 emissions from palsas (this issue is most often studied through computer modeling, with a focus on air or soil temperature). In this article, the authors attempt to fill this knowledge gap, which is undoubtedly important. The language is clear and precise, and the article is well structured and easy to follow. There are some questions regarding the conclusions, which could perhaps be made more specific, as well as some questions concerning the methodology, which will be discussed further. Overall, with minor revisions, I recommend the article for publication.

**Specific comments:**

1. Please clarify how the authors distinguished the effect of soil temperature from (i) the direct physical process of thawing and (ii) the effect of flooded. If the intention is to assess specifically the influence of flooded (or soil moisture), this effect should be explicitly separated from the permafrost-thawing process. In particular, please state in Section 2.3.3 the time interval between the thawing stage and the addition of water. If this interval was short, CO2 emission may have been driven predominantly by the rise in soil temperature and by the physical release of CO2 previously trapped in permafrost, which would complicate attribution of the observed CO2 emissions to soil flooding.

AR: We appreciate the reviewer's comment and agree that separating the effects of soil temperature, physical thawing, and flooding is important for interpreting CO2 fluxes. Our experimental setup consisted of a control sample (non-flooded) and three different flooded treatments, which allowed us to assess the effect of flooding. However, the aim of this study was not to isolate the effect of flooding alone, but rather to investigate the effect of flooding during permafrost thaw. This explains why we did not include a flooded treatment without sequential thawing. In section 2.3.3, we will specify that we thawed the soil columns overnight in the incubation chamber and subsequently added water to simulate in-situ abrupt thaw.

"We thawed the samples overnight, and added water the day after."

As mentioned in line 406, we agree that sequential thawing can amplify CO2 pulses following permafrost thaw due to physical disruption. Since we thawed both the control and flooded mesocosms simultaneously under similar conditions, the release of CO2 trapped in permafrost should have been comparable across all mesocosms, allowing differences to be attributed primarily to the flooding component.

We will include this clarification and clarify the limitation of the experimental setup regarding potential disturbances caused by the physical process of thawing in section 4.2; 1.406.

**Suggestion:**

" $CO_2$  pulses following thaw could also be related to releases of trapped  $CO_2$ . Finally, these  $CO_2$  pulses may also be enhanced by physical disturbances due to sequential thawing and, therefore, may not represent natural processes. However, since all the mesocosms were thawed under similar conditions, potential releases of trapped  $CO_2$  or physical disturbances due to setup limitations should be comparable across mesocosms (excluding micro-scale spatial heterogeneity)."

2. In the context of your study, it would be valuable to measure CO2 efflux (soil respiration) in situ at both the fen and the palsa to determine whether the differences observed in the laboratory mesocosms are reproduced under field conditions. The literature reports diverse findings: some studies find higher CO2 efflux from palsas than from fens due to drier conditions in palsas and the suppressive effect of anaerobic conditions in fens, whereas other studies report equal or greater CO2 emissions from fens. Because mesocosm experiments that closely approximate natural conditions are relatively scarce, it would be particularly informative to assess the correspondence between your laboratory mesocosm results and field measurements.

AR: Thank you for the remarks. We did measure in-situ CO2 emissions at both sites and found higher CO2 respiration rates (in ppm) at the fen compared to the palsa. We also measured CH4 emissions at the fen. Although we agree that including the in-situ measurements would be valuable; we chose not to include them as our data are based on a single field campaign. As shown by Voigt et al. 2017, soil respiration varies throughout the year and therefore, our limited temporal coverage may not provide sufficiently representative results.

3. Why were samples collected in winter? In this case, how long does it take for CO2 efflux to stabilize after thawing the samples? The thawing process strongly affects CO2 emissions; although a 12-week stabilization period is likely sufficient in many cases, did you test stabilization time experimentally?

AR: We collected the samples in winter to avoid soil compactions during the coring and microbial disruption due to the freezing (l. 106-107). To allow  $CO_2$  fluxes to stabilize between each thawing steps, we had an incubation of four weeks for between each thaw disturbance. We determined the  $CO_2$  stabilization period during the first thawing step (active layer) and kept the same for all the incubation time. We will clarify this in the method section.

Suggestion 1. 153: "To thaw the permafrost, we deepened the active layer by 20 cm every four weeks (Table S2). The duration of each sequential thaw was determined based on the time need for  $CO_2$  fluxes to stabilize during the first thaw thaw step. For consistency, we applied the same thaw duration for each thaw step. However, we acknowledge that the duration for  $CO_2$  fluxes to stabilize following thaw might have differed for each sequential thaw."

4. The manuscript would benefit from a clearer discussion of the temporal distinction between abrupt thaw and gradual thaw.

AR: Following the response from Reviewer 1, we will redefine clearly the difference between abrupt and gradual thaw based on the review paper of Webb et al. (2025). In this manuscript, we focus mainly on the short-term CO2 responses due hydrological differences between the two types of thaws rather than the temporal aspect. However, we agree that we a paragraph on the temporal perspective would improve the manuscript. We will add a short paragraph in section 4.2. focusing on the temporal distinctions between the two types of thawing and implications for C emissions.

5. Why was a peat sample from a fen affected by gradual thaw used as the control (Figure 2)? It seems more appropriate to use a palsa sample unaffected by thaw as the control, and then to compare it with the effects of gradual and abrupt thaw. Please explain the choice of control.

AR: In this setup, we used the fen mesocosm as an end-member to compare long-term CO2 and CH4 emissions with those from the short-term simulated thaw (palsa samples), as described in Section 2.3.1 (lines 126–127). We expected CH4 emissions from the fen site due to its longer permafrost-free conditions and a distinct microbial community. We further clarified this now by stating:

"A core from the fen was included as an end-member representing long-term thawed conditions (~60 years). This allowed comparison of CO2 and CH4 emissions following short-term simulated permafrost thaw with those from a naturally thawed fen."

6. The conclusion implies CO2 transport via plant aerenchyma; however, aerenchymatous tissues primarily serve as conduits for CH4 transport from anoxic peat to the atmosphere, bypassing aerobic, methane-oxidizing layers. This process can lead to reduced CO2 emissions in fens with vascular plants, since it limits the oxidation of CH4 to CO2 (Lai, 2009).

AR: We agree that aerenchyma tissue facilitates CH4 transport and that differences between dry and wet environments can influence CO2 emissions. We will clarify this point by: (1) rephrasing the conclusion to make it clear that this is presented as a hypothesis rather than a definitive statement, and (2) specifying in section 4.3 that although aerenchyma tissue enables CH4 transport and reduces CH4 oxidation, it can also serve as a pathway for CO2 transport. This is because CO2 diffuses approximately 10,000 times faster in air than in water (Armstrong 1980; Colmer 2003; Tiner 2005).

**The text will read:**

- 1. "Additionally, CO2 emissions from fens increased under water-saturated conditions (H3), possibly reflecting CO2 transport from deeper soil layers mediated by deep-rooted vegetation during non-growing and shoulder seasons."
- 2. "While surface vegetation was removed during incubation, the root systems were retained. The presence of deep-rooted aerenchymous plants, such as sedges, may explain the increased CO2 emissions in the fen. Aerenchyma tissue enhances the transport of O2, CO2, and CH4 in waterlogged soils (Armstrong, 1980; Colmer, 2003). Although aerenchyma are best known for enhancing CH4 transport and reducing CH4 oxidation, they can also provide pathways for CO2 movement. Because CO2 diffuses roughly 10,000 times faster in air than in water (Armstrong, 1980; Colmer, 2003; Tiner, 2005), gas exchange through aerenchyma can substantially influence CO2 fluxes. In this process, O2 is transported downward to the roots, while CO2 produced by microbial respiration moves upward and may be fixed in the shoots via photosynthesis (Dacey and Klug, 1982; Pedersen and Sand-Jensen, 1992; Smith and Russell, 1969; Smith et al., 1983)."

I thank the authors for this interesting article, which has also provided me with valuable insights relevant to my own field of research:)

AR: You are very welcome, and we appreciate your feedback and comments to help us further improve this manuscript.